# Automated discovery of a robust interatomic potential for aluminum

Justin S. Smith [1,2 ✉], Benjamin Nebgen [1 ✉], Nithin Mathew[1,2], Jie Chen[3], Nicholas Lubbers[4], Leonid Burakovsky [1], Sergei Tretiak [1], Hai Ah Nam [4], Timothy Germann [1], Saryu Fensin[3] & Kipton Barros[1 ✉]

Machine learning, trained on quantum mechanics (QM) calculations, is a powerful tool for modeling potential energy surfaces. A critical factor is the quality and diversity of the training dataset. Here we present a highly automated approach to dataset construction and demonstrate the method by building a potential for elemental aluminum (ANI-Al). In our active learning scheme, the ML potential under development is used to drive non-equilibrium molecular dynamics simulations with time-varying applied temperatures. Whenever a configuration is reached for which the ML uncertainty is large, new QM data is collected. The ML model is periodically retrained on all available QM data. The final ANI-Al potential makes very accurate predictions of radial distribution function in melt, liquid-solid coexistence curve, and crystal properties such as defect energies and barriers. We perform a 1.3M atom shock simulation and show that ANI-Al force predictions shine in their agreement with new reference DFT calculations.

[1] Theoretical Division, Los Alamos National Laboratory, Los Alamos, NM, USA. [2] Center for Nonlinear Studies, Los Alamos National Laboratory, Los Alamos, NM, USA. [3] Materials Division, Los Alamos National Laboratory, Los Alamos, NM, USA. [4] Computer, Computational, and Statistical Sciences Division, Los Alamos National Laboratory, Los Alamos, NM, USA. ✉email: just@lanl.gov; bnebgen@lanl.gov; kbarros@lanl.gov

Given sufficient training data, ML models show great promise to accelerate scientific simulation, e.g., by emulating expensive computations at a high accuracy but much reduced computational cost. ML modeling of atomic-scale physics is a particularly exciting area of development[1–4]. Provided sufficient training data, ML models suggest the possibility for the development of models with unprecedented transferability. Applications to materials physics, chemistry, and biology are innumerable. To give some examples, simulations for crystal structure prediction, drug development, materials aging, and high strain/strain-rate deformation would all benefit from better interatomic potentials.

Machine learning (ML) of interatomic potentials is a rapidly advancing topic, for both materials physics[5–18] and chemistry[19–26]. Training datasets are calculated from computationally expensive ab initio quantum mechanics methods, most commonly density functional theory (DFT). Trained on this data, an ML model can be very successful in predicting energy and forces for new atomic configurations. ML potentials typically assume very little beyond symmetry constraints (e.g., translation and rotation invariance) and spatial locality (each atomic force only depends on neighboring atoms within a fixed radius, typically of order 5–10 Å). Long-range Coulomb interactions or dispersion corrections may also be added[21,27].

For large-scale molecular dynamics (MD) simulations, so-called classical potentials are usually the tool of choice. Such potentials are relatively simple and computationally efficient. Although effective for many purposes, classical potentials may struggle to achieve broad transferability. For example, it is not easy to design a single classical potential that correctly describes multiple incompatible crystal phases and the transitions between them. Consequently, assumed functional forms for classical potentials tend to grow more flexible over time. For example, the embedded atom method (EAM)[28] has lead to generalizations such as modified EAM (MEAM)[29] and multistate MEAM[30].

In contrast to classical potentials, the ML philosophy is to begin with a functional form of the utmost flexibility. For example, a modern neural network-based ML potential may contain ~$10^5$ fitting parameters. If properly trained, recent work suggests that the accuracy of ML potentials can approach that of the underlying ab initio theory (e.g., DFT or coupled cluster)[4,20,21,25,31–34]. Although slower than classical potentials, ML potentials are vastly faster than, say, reference DFT calculations. The main limitation on the accuracy and transferability of an ML potential is the quality and broadness of the training dataset.

In this paper, we design an active learning approach for automated dataset construction suitable for materials physics and demonstrate its power by building a robust potential for aluminum that we call ANI-Al. Distinct from previous works, here the active learning scheme receives very limited expert guidance. In particular, we do not seed the training dataset with any crystal or defect structures; the active learning scheme begins only with fully randomized atomic configurations. By leaving the search space of possibly relevant atomic configurations unspecified, we aim to build a model that is maximally general. If successful, the model should remain accurate when presented with complex atomic configurations that may arise in a variety of highly nonequilibrium dynamics.

The basic steps of active learning (AL) for atomic-scale modeling are to sample new atomic configurations, query the ML model for uncertainty in its predictions, and selectively collect new training data that would best improve the model[24,35–40]. Previous work employed AL to drive nonequilibrium sampling of large datasets through organic chemical space, yielding the highly general ANI-1x potential[41]. Other recent research by Gubaev et al.[42] has explored the use of AL with moment tensor potentials to construct atomistic potentials for materials. Zhang et al. also applied AL to materials using the deep potential model[32] for MgAl alloys. AL was used by Deringer, Pickard, and Csányi to build an accurate and general model for elemental Boron[43].

The AL procedure developed in this work will be discussed in detail below, but briefly, there is a loop over three main steps: (1) using the best ANI-Al models available, MD simulations with time-varying temperatures are performed to sample new atomic configurations; (2) an ML uncertainty measure determines whether the sampled configurations would be useful for inclusion in the training data and if so, new DFT calculations are run; and (3) new ANI-Al models are trained with all available training data. The starting point for AL is an initial training dataset consisting of DFT calculations on randomized (disordered) atomic configurations. Each MD sampling trajectory is also initialized to a random disordered configuration, with random density. Required human inputs to the active learning procedure include the range of temperatures and densities over which to sample and various ML hyperparameters that are largely transferable between materials. Crucially, the AL scheme receives no a priori guidance about the relevant configuration space it should sample. Nonetheless, after enough iterations, the AL procedure eventually encounters configurations that locally capture characteristics of crystals such as FCC, HCP, BCC, and many others. The AL algorithm is readily parallelizable; we employed hundreds of nodes on the Sierra supercomputer to collect the final ANI-Al dataset consisting of about 6000 DFT calculations.

We demonstrate, via a large set of benchmarks, that the resulting ANI-Al potential is effective in predicting many properties of aluminum in liquid and crystal phases. The performance on crystal benchmarks is notable, given that the automatically generated AL training dataset consists primarily of disordered and partially ordered configurations. ANI-Al shines when applied to extreme and highly nonequilibrium processes. As a test, we perform a 1.3 M atom shock simulation and verify the ANI-Al-predicted forces by performing new DFT calculations on randomly sampled local atomic environments. Force prediction errors (per component) are of order 0.03 eV/Å, whereas typical force magnitudes ranged from 1 to 2.5 eV/Å. In terms of absolute force accuracy, ANI-Al performs nearly as well for extreme shock simulations as it does for equilibrium crystal or liquid simulations. To help understand the impressive transferability of ANI-Al, we present a two-dimensional visualization of the space of configurations sampled in the AL training dataset. The liquid phase, a variety of crystal structures, and the highly defected configurations that appear in shock all appear to be well-sampled.

## Results

Here, we present a variety of benchmarks for ANI-Al, our machine-learned potential for aluminum. First, we report crystal property predictions, including energies, elastic constants, energy barriers, phonon spectrum, point defect energies, and surface properties. Next, we present results on the liquid phase and on transitions between liquid and crystal. Our final application is a large-scale shock simulation, for which we verify ML-predicted forces using new DFT calculations. Finally, we illustrate the advantages of the AL approach by characterizing the diversity of configurations sampled.

**Predicting crystal energies.** Figure 1 shows ANI-Al-predicted energies (solid lines) for select crystal structures. ANI-Al correctly predicts that FCC has the lowest energy of all crystals considered; more crystal energies are compared in Supplementary Table 5. Vertical bars show the sample variance over the eight neural

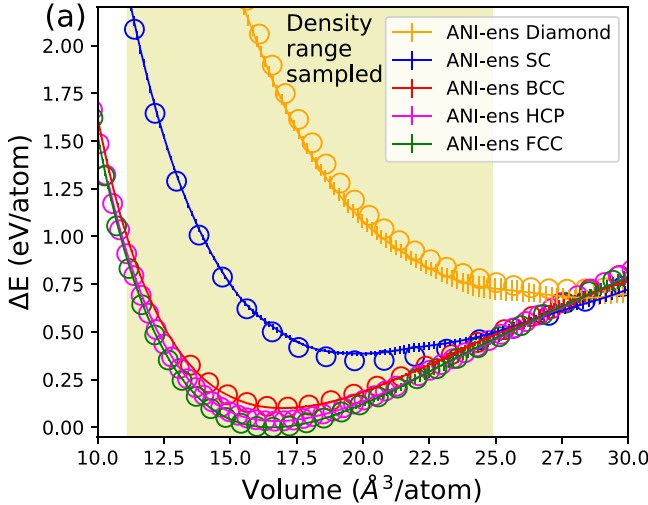

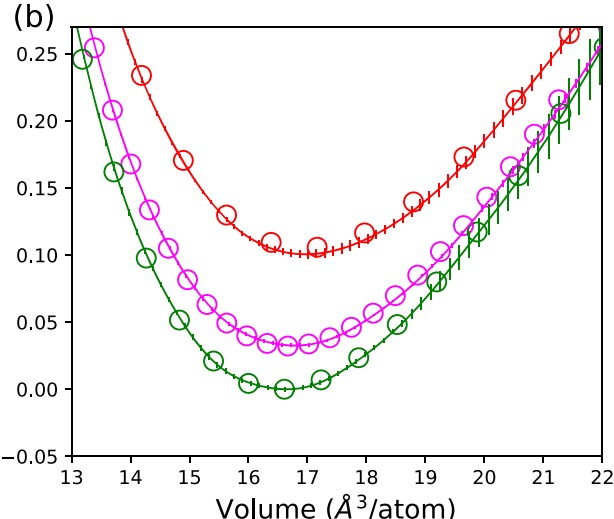

**Fig. 1 Crystal energies relative to the ground state.** Solid lines represent ANI-Al predictions and circles represent density functional theory (DFT) reference calculations. Vertical bars represent sample variance of the eight neural networks comprising the (ensembled) ANI-Al model. Panel (**b**) is a magnification of panel (**a**) near the energy minima. The highlighted yellow region (~11–25 Å³/atom) indicates the approximate range of densities sampled in the training data. Crystal structures are diamond, simple cubic (SC), body-centered cubic (BCC), hexagonal close-packed (HCP), and face-centered cubic (FCC).

networks that comprise a single ANI-Al model (i.e., the uncertainty measure used within the active learning procedure). DFT reference data are shown in circles.

For both ANI-Al and DFT calculations, energies are measured relative to the FCC ground state. Let $\epsilon_x$ represent the error of the ANI-Al predicted energy for crystal structure $x$ at its energy minimizing volume (volume is independently optimized for ANI-Al and DFT). By definition, the energy shifts are such that $\epsilon_{\text{fcc}} = 0$. After FCC, the second-lowest energy structure shown in this plot is HCP, for which the ANI-Al error is $\epsilon_{\text{hcp}} = 0.42$ meV/atom. Note that FCC and HCP are competing close-packed structures, and both can reasonably be expected to emerge in our active learning dynamics (FCC with a stacking fault looks locally like HCP). BCC, by contrast, is only physical in aluminum at much higher densities, far beyond the range of our active learning sampling. It is not surprising, therefore, that the ANI-Al error for

BCC is an order of magnitude larger, $\epsilon_{\text{bcc}} = 5.3$ meV/atom. Simple cubic and diamond crystals are less physical still, and we observe $\epsilon_{\text{sc}} = 37$ meV/atom and $\epsilon_{\text{diamond}} = -44$ meV/atom. Nonetheless, the qualitative agreement between ANI-Al and DFT observed in Fig. 1, even for very unphysical crystals, seems remarkable. Similar observations were made in ref. [32]. We emphasize that in the present work, the training data include no hand-selected crystals. Instead, all atomic configurations in the training dataset were generated using MD sampling, starting only from disordered configurations.

ANI-Al predictions are most reliable for the range of densities sampled in the training data (Fig. 1a, yellow region). A further extrapolation of these cold curves is shown in Supplementary Fig. 9.

**Predicting elastic constants**. We can compare ANI-Al-predicted elastic constants against experimental data. A particularly important one is the bulk modulus, which corresponds to the curvature of the FCC cold curve at its minimum (Fig. 1b). Experimentally, the FCC bulk modulus is measured to be 79 GPa [44], whereas the ANI-Al prediction is 77.3 GPa. The full set of FCC elastic constants is measured experimentally to be, $C_{11} = 114$ GPa, $C_{12} = 61.9$ GPa, and $C_{44} = 31.6$ GPa[44]. For our DFT calculations, $C_{11} = 106$ GPa, $C_{12} = 62.3$ GPa, and $C_{44} = 31.6$ GPa. For ANI-Al, we predict $C_{11} = 117$ GPa, $C_{12} = 57.2$ GPa, and $C_{44} = 30.4$ GPa.

The largest discrepancies between ANI-Al and DFT are observed for the elastic constants $C_{11}$ and $C_{12}$, with relative errors of 10.38% and $-8.19\%$, respectively. Interestingly, the effects of these two discrepancies seem to cancel in the bulk modulus, $B = (1/3)(C_{11} + 2C_{12})$, for which the error relative to DFT is just 0.78%. We suspect the cancellation is not a coincidence, because a similar phenomenon was observed in previous ML potentials developed for aluminum[32,34] (cf. Supplementary Table 4). Elastic constants measure the response of stress to a small applied strain. For an ML model to precisely capture $C_{ij}$, its training data should ideally contain many locally perfect FCC configurations for a variety of small strains. The mechanisms by which our active learning sampler can generate strained FCC are somewhat limited (e.g., nucleation of imperfect crystals). Future work might employ time-varying applied strains to the entire supercell, in addition to the time-varying temperatures employed in this study.

In predicting elastic constants, ANI-Al accuracy is on par with many classical potentials and existing ML potentials, as shown in Supplementary Tables 3 and 4. Whereas classical potentials are usually designed to reproduce experimental elastic constants, in ANI-Al this capability is an emergent property. Our active learning sampling discovers the FCC lattice and its properties in an automated way.

**Predicting crystal energy barriers**. The Bain path (Fig. 2a) represents a volume-preserving homogeneous deformation that transforms between FCC and BCC crystals. Starting from the initial FCC cell ($c/a = 1$), we compress along with one of the $\langle 100 \rangle$ directions (length $c$) while expanding equally in the two orthogonal directions (lengths $a = b$). The special value of $c/a = 1/\sqrt{2} \approx 0.71$ would correspond to BCC symmetry. Figure 2a shows energies along this Bain path, in which $c/a$ varies continuously while conserving volume, $a^2c$. The observed maximum at $c/a = 1/\sqrt{2}$ indicates that the BCC structure is unstable to tetragonal deformation. We compare ANI-Al to DFT reference calculations, as well as seven EAM-based potentials[45–52]. Supplementary Fig. 2 quantifies the errors for each potential, averaged over the strain path.

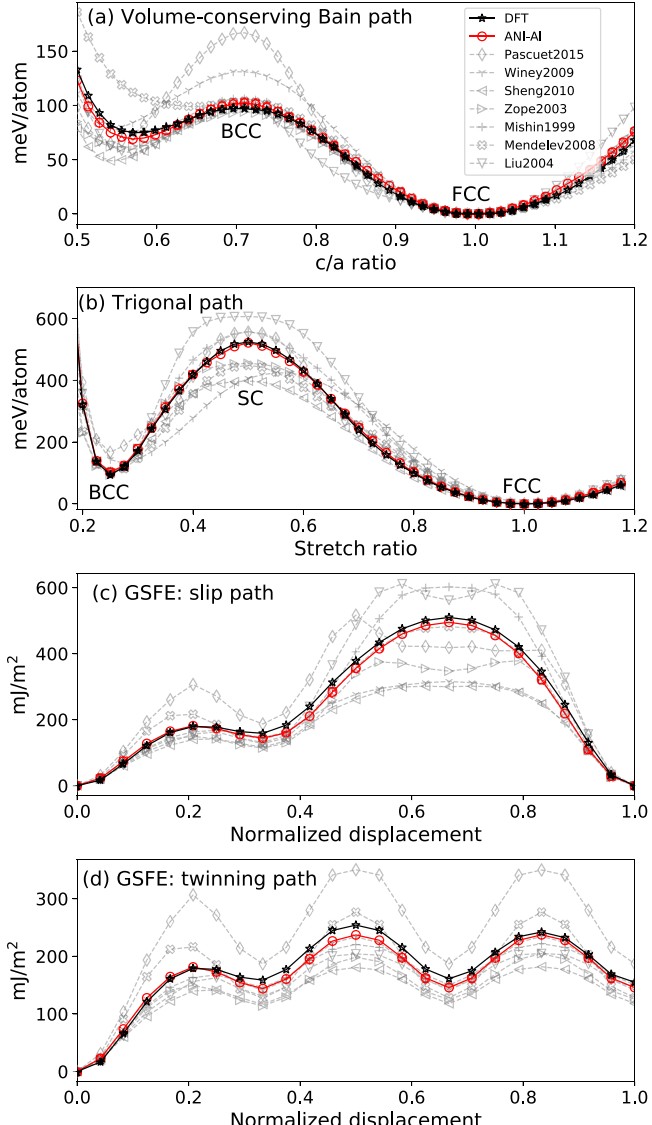

**Fig. 2 Transformational energy barriers.** We compare ANI-Al and various classical potentials to reference DFT data. **a** Volume-conserving Bain path energies. **b** Trigonal path energies. **c** Generalized stacking fault energy (GSFE) slip path. **d** GSFE twinning path.

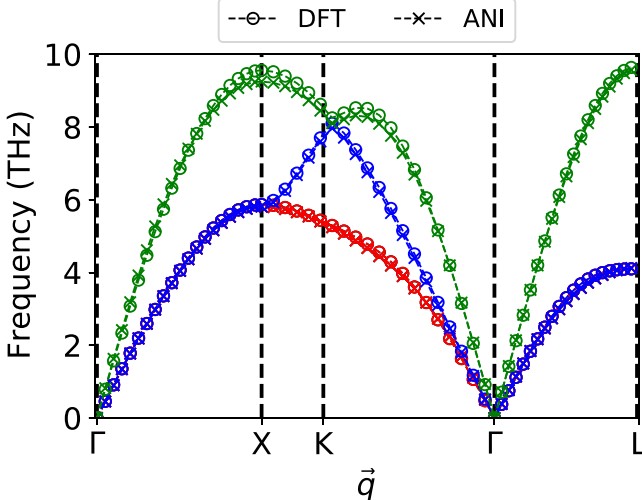

**Fig. 3 Comparison of predicted vs. reference phonon spectrum.** Phonon spectrum of FCC Al predicted by the ANI-Al model (crosses) and compared to the DFT reference (circles).

and the energy per unit area required to form a single stacking fault. The GSFE twinning path (also known as the generalized planar fault energy) is an extension of the slip path and provides an estimate of the energy per unit area required to form $n$-layer faults (twins) by shearing $n$ successive {111} layers along ⟨112⟩. We calculated the GSFE slip path and the twinning paths using standard methods[53–56].

Figure 2c, d shows energies along with the GSFE slip and twin paths, respectively. As before, we compare with seven EAM-based potentials. The ANI-Al potential agrees quite well with the reference DFT data for all measurements in Fig. 2. To quantify this agreement, we calculate the root-mean-squared error (RMSE), formed as an average over the Bain, Trigonal, GFSE slip, and GFSE twinning paths. ANI-Al achieves RMSE values of 4.5 meV/atom, 6.0 meV/atom, 16.6 mJ/m$^2$, and 11.4 mJ/m$^2$, respectively. For predicting these paths, the best classical potential is by Mishin et al.[48], which achieves errors of 4.3 meV/atom, 37.6 meV/atom, 52.5 mJ/m$^2$, and 15.9 mJ/m$^2$. Supplementary Fig. 2 quantifies the errors for each potential, averaged over the strain path. It is interesting to note that the Winey et al. potential[50], which does exceptionally well in predicting many FCC properties (see Supplementary Table 3), struggles to accurately predict the Bain and GSFE slip paths.

Figure 2b shows the energies along the trigonal deformation path, where the ideal FCC phase is compressed along the $z = $ ⟨111⟩ crystallographic direction, and elongated equally in the two orthogonal directions $x$ and $y$, such that the total volume is conserved. We define a characteristic "stretch ratio" as $(L'_z/L'_x)/(L_z/L_x)$, where $L_z$ and $L_x$ are the dimensions of the reference FCC simulation cell along $z$ and $x$ directions, respectively, and $L'_z$ and $L'_x$ are the dimensions of the deformed simulation cell. Stretch ratios of 1.0, 0.5, and 0.25 result in FCC, SC, and BCC phases, respectively. Good agreement is found between ANI-Al and DFT reference calculations. It can be seen from Fig. 2b that SC, but not BCC, is unstable to trigonal deformation.

A stacking fault in FCC represents a planar defect in which the crystal locally is in HCP configuration within the nearest-neighbor shell (note that FCC and HCP are competing close-packed structures). The generalized stacking fault energy (GSFE) slip path provides an estimate of the resistance for dislocation slip

**Predicting FCC phonon spectrum.** Figure 3 compares the ANI-Al predicted phonon spectrum to that of DFT. In both cases, the frequencies were calculated using the PHON program[57] via the small-displacement method[58,59]. A supercell of size $4 \times 4 \times 4$ FCC unit cells was used for the calculations. The ion at the origin of this supercell was displaced in [100], with a magnitude of 1% the equilibrium FCC lattice spacing, and the forces were calculated on all the ions. These forces were used to calculate the phonon frequencies in the quasi-harmonic approximation. Figure 3 shows good agreement between ANI and DFT predictions of the FCC Al phonon spectrum.

**Predicting FCC point defects.** ANI-Al predicts the formation energies for vacancy and (⟨100⟩ dumbbell) interstitial defects to be 663 meV and 2.49 eV, respectively. The corresponding DFT predictions are 618 meV and 2.85 eV. The vacancy formation energy is experimentally estimated to be ~680 meV[60].

Supplementary Tables 3 and 4 also list predictions for existing classical and ML potentials. The relatively large deviation between ANI-Al and DFT predictions is perhaps an indication that vacancies and interstitials did not play a large role in the configurations sampled during the active learning procedure.

**Predicting FCC surface properties.** The properties of surfaces predicted by our ANI-Al model is compared to values from DFT, experiments, and seven EAM-based potentials in Supplementary Table 3. Supplementary Table 4 compares with previous ML results, where available. ANI slightly overpredicts the surface energy for {100}, {010}, and {111}, with a maximum error of 6.6% (for {100}) compared to DFT predictions. ANI predicts the correct sign for surface relaxation (inward or outward) in all but one case ($d_{12}^{\{100\}}$). The outward relaxation of {100} and {111} surfaces in Al are considered "anomalous" and ANI predicts this correctly only for {111}, despite correct predictions by DFT for both surfaces. Also note that ANI-Al correctly predicts the ordering of the magnitudes of surface relaxation, $|d_{12}^{\{110\}}|>|d_{12}^{\{100\}}| \approx |d_{12}^{\{111\}}|$, but the quantitative agreement with DFT reference calculations is poor. The ANI-Al training dataset includes only bulk systems with periodic boundary conditions, but some surface configurations may have been incidentally sampled due to void formation at low densities.

**Predicting radial distribution functions.** To validate our ANI-Al model in the liquid phase, we carry out MD simulations to measure radial distribution functions (RDF) and densities at various temperatures. Figure 4a compares simulated RDFs with experimental measurements at 1123, 1183, and 1273 K[61]. Independent simulations were performed in the isobaric–isothermal (NPT) ensemble to determine equilibrium densities of liquid Al at the relevant (P,T) conditions. MD simulations of 2048 atoms were initialized at these densities and equilibrated for 50 ps in the NVT ensemble using the Nosé–Hoover-style equations of motion[62] derived by Shinoda et al.[63] Reported RDFs were calculated (bin size of 0.05 Å) by averaging 100 instantaneous RDFs, which were 0.1 ps apart, in the final 10 ps of the NVT equilibration. A timestep of 1 fs was used for these simulations. Figure 4b compares ANI-Al predicted densities at various temperatures (still at atmospheric pressure) to multiple experimental values[64–69]. For reference, the melting temperature is $T_{melt} \approx 933$ K. The agreement between ANI-Al predictions and experiment is comparable to the variation between different experiments.

**Predicting liquid–solid phase boundaries.** Figure 5 shows the liquid–solid coexistence line in the pressure–temperature plane. At each pressure, we calculated the coexistence temperature by performing simulations with an explicit solid–liquid interface[70–72]. The details of these simulations are provided in Supplementary Note 7. Experimental data are available up to about 100 GPa[73]. We also compare with prior DFT calculations[46] and a classical MD potential. For the latter, we used the Mendelev et al. potential[45], which was explicitly parameterized to model the melting point of aluminum, $T_{melt} \approx 933$ K at atmospheric pressure. At this pressure, both Mendelev and ANI-Al potentials predict an FCC melting point of ~925 K, in good agreement with the experiment.

The Mendeleev model begins to underestimate the melting temperature at around 5 GPa, whereas the ANI-Al model remains quite accurate up to ~50 GPa. Note that the ANI-Al training data were restricted to a limited range of densities (yellow region of Fig. 1a) which correspond to pressures up to ~50 GPa (See Supplementary Fig. 1). We were surprised to observe qualitative agreement between the ANI-Al and DFT predicted coexistence curves up to 250 GPa, even though this is a significant extrapolation for ANI-Al.

For the Mendelev simulations, the liquid–FCC coexistence curve only extends to ~20 GPa; beyond that point, we observed nucleation into BCC. According to prior DFT-based studies[46,74] and experiment[75], the solid-to-solid transition out of FCC should require hundreds of GPa. Figure 5 includes the theoretically predicted liquid–BCC coexistence curve at pressures between 225 to 275 GPa.

**Phase-transition dynamics.** Next, we carry out a nonequilibrium MD simulation to observe both freezing and melting dynamics. Our intent is to verify the ANI-Al-predicted energies and forces at snapshots along the dynamical trajectory. Along the trajectory the temperature is slowly increased from 300 to 1500 K, then cooled back to 300 K. The details of these simulations are provided in Supplementary Note 7.

Figure 6 shows the potential energy, mean force magnitude, and pressure for both ANI-Al and DFT along this trajectory. Melting from FCC to liquid occurs at around 300 ps, and freezing occurs at around 700 ps. The pressure was calculated using the method of ref. 76. The inset images in Fig. 6b show the composition of the system before and after melting, and after refreezing. Compositional information was obtained using the common neighbor analysis as implemented in the OVITO visualization software[77].

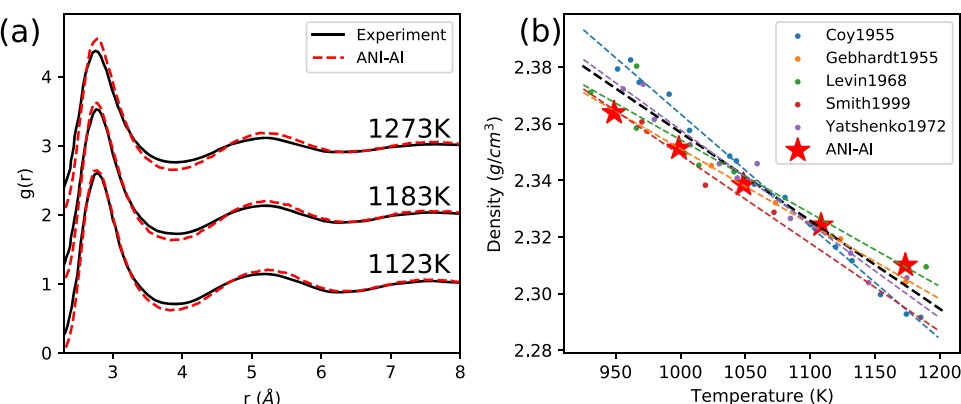

**Fig. 4 Molecular dynamics simulation in melt using the ANI-Al potential. a** Radial distribution function at temperatures 1123, 1183, and 1273 K compared to experiment[61] (black line). **b** Density predictions as a function of temperature. The dashed black line is a linear fit to all five sources of experimental data.

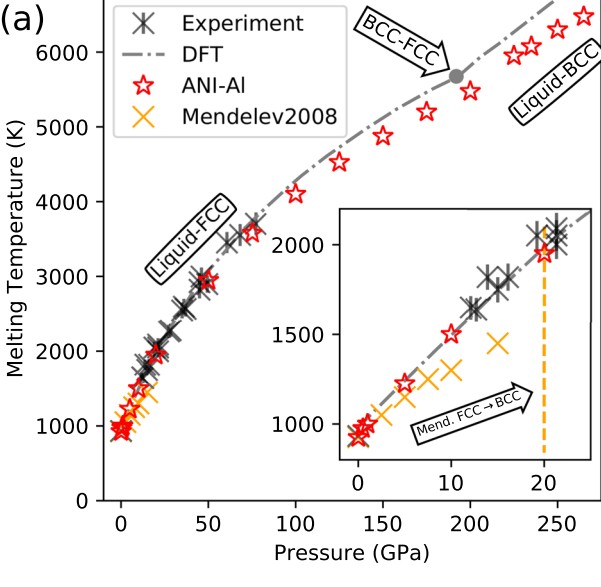

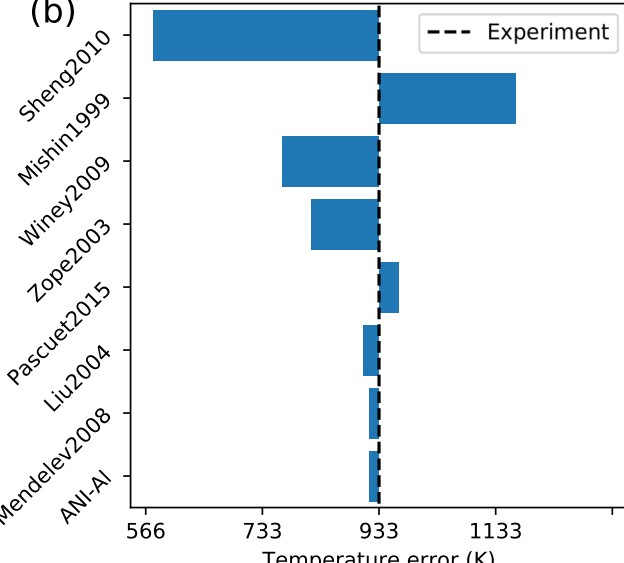

**Fig. 5 Predicting melt temperatures. a** Melt curve as a function of pressure for DFT[46], ANI-Al, and the Mendelev et al. EAM potential[45], compared with experimental data[73]. Below 210 GPa, we show FCC–liquid coexistence. Above 210 GPa, we show BCC–liquid coexistence. The inset zooms to pressures from 0 to 20 GPa. **b** Errors in predicting the melt temperature at atmospheric pressure.

Every 2.5 ps along the trajectory we sampled a frame to perform reference DFT calculations. The error between ANI-Al and DFT is generally small. Averaged over the full trajectory, the MAE for energy is 0.84 meV/atom. The MAE for each force component individually is 0.023 eV/Å. The MAE for ANI-Al predicted pressure is 0.36 GPa. Interestingly, there is a tendency for ANI-Al to overestimate pressure, especially at negative pressures. Perhaps this systematic error reflects the fact that a large fraction of the ANI-Al training data was sampled at very large positive pressures (cf. Supplementary Fig. 1). Model performance in predicting pressure could likely be significantly improved by including pressure data in the training procedure.

Supplementary Figs. 3 and 4 further verify the ANI-Al force predictions for MD simulations over large a range of temperatures and densities.

**Simulation of shock physics.** Finally, to verify our potential at predicting material response under extreme conditions, we carried out a large-scale shock simulation using NeuroChem interfaced to the LAMMPS molecular dynamics software package[78]. The simulation cell, containing about 1.3 M atoms, has approximate dimensions $10 \times 211 \times 10$ nm in the $x = [112]$, $y = [\bar{1}10]$, and $z = [\bar{1}\bar{1}1]$ crystallographic directions. Prior to shock, the volume was equilibrated at 300 K for 15 ps in the *NVT* ensemble. Periodic boundary conditions were applied in $x$ and $z$, with free surfaces in $y$. After equilibration, a $u_p = 1.5$ km/s shock was applied in $y$ using the reverse-ballistic configuration[79], and the system was evolved in the *NVE* ensemble. In this method, a rigid piston is defined by freezing a rectangular block of atoms and the velocities of the remaining atoms are modified by adding $-u_p$ to the $y$ component. This sets up a supported shock wave in the flexible region of the simulation cell. Using spatial domain decomposition as implemented in LAMMPS, the 1.3 M atoms were distributed across 80 Nvidia Titan V GPUs, and the required wall-clock time for the entire 31 ps simulation (62 k MD timesteps) was about 15 h.

Figure 7a shows the dislocation structure in the simulation cell, as predicted by the Dislocation Extraction Algorithm (DXA)[77,80] at 24.5 ps.

We randomly selected five atoms in the simulation volume for further analysis. The atomic environments for these five atoms are shown as clusters and highlighted with colored boxes in Fig. 7a. The five zoomed insets illustrate that dislocations can pass near each of the five central atoms at specific times, which are marked with colored boxes in Fig. 7b–f.

Figure 7b–f compares the ANI-Al predicted forces with new reference DFT calculations at every 0.5 ps of simulation time. For each sample point, a local environment (a cluster of radius 7 Å) was extracted from the large-scale shock simulation and placed in a vacuum. A new DFT calculation was performed on this cluster, and the resulting force on the central atom was compared to the corresponding ANI-Al prediction. As shown in Fig. 7b–f, the magnitudes of the forces have a characteristic scale of order 1 eV/Å. The mean absolute error, for the ANI-Al predictions of each force component individually, is ~0.06 eV/Å. However, as we will discuss below, artificial surface effects due to finite cluster radius $r = 7$ Å cannot be neglected, and larger clusters are required to measure the true ANI-Al error for these shock simulations.

To systematically study the effect of cluster cutoff radius $r$, we further down-sampled to ten local atomic environments. Figure 7g quantifies the $r$-dependence on the DFT-calculated force $\mathbf{f}_r$. Specifically, it shows the mean of $|f_{r;a} - f_{r_0;a}|$, where the reference radius is taken to be $r_0 = 10$ Å. Averages were taken over all force components $a = x, y, z$ and over all ten local atomic environments. Surface effects for $r = 7$ Å are seen to modify the central atom force by about 0.06 eV/Å, which is of the same order as the ANI-Al disagreement with DFT, when measured using this $r$. The average force magnitude for these ten configuration samples is 1.12 eV/Å, so the observed deviations at $r = 7$ Å represent about a 5% effect.

Figure 7h illustrates that ANI-Al and DFT agreement becomes better for calculations on larger clusters, i.e., where artificial surface effects are reduced. With cluster radius $r = 7$ Å, the ANI-Al mean absolute error (MAE) for force components is about 0.06 eV/Å. At the largest cluster size, we could reach ($r = 10$ Å) the ANI-Al MAE reduces to ~0.03 eV/Å, i.e., about a 3% relative error. For reference, recall that the ANI-Al force errors in the section "Phase transition dynamics" were slightly lower, at 0.023 eV/Å; in that context, however, the reference DFT calculations did not suffer from artificial surface effects.

It makes sense that ANI-Al and DFT forces are most consistent for the largest cluster sizes, given that the training data produced by active learning consists entirely of bulk systems. Note that

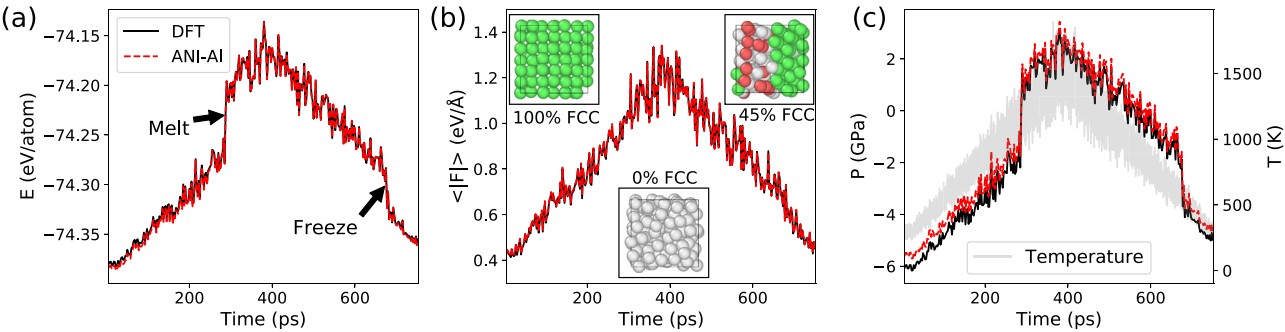

**Fig. 6 ML-driven molecular dynamics, showing melting and freezing processes.** The system is heated from 300 to 1500 K, and cooled back to 300 K. Reference DFT calculations (black) are used to verify the ANI-Al predictions (red) for (**a**) the energy, (**b**) mean (avg.) force, and (**c**) pressure. The instantaneous temperature is shown in gray on panel (**c**). Panel (**b**) insets show the local atomic structure (green—FCC; gray—disordered; red—HCP) at snapshots before melting, after melting, and after refreezing.

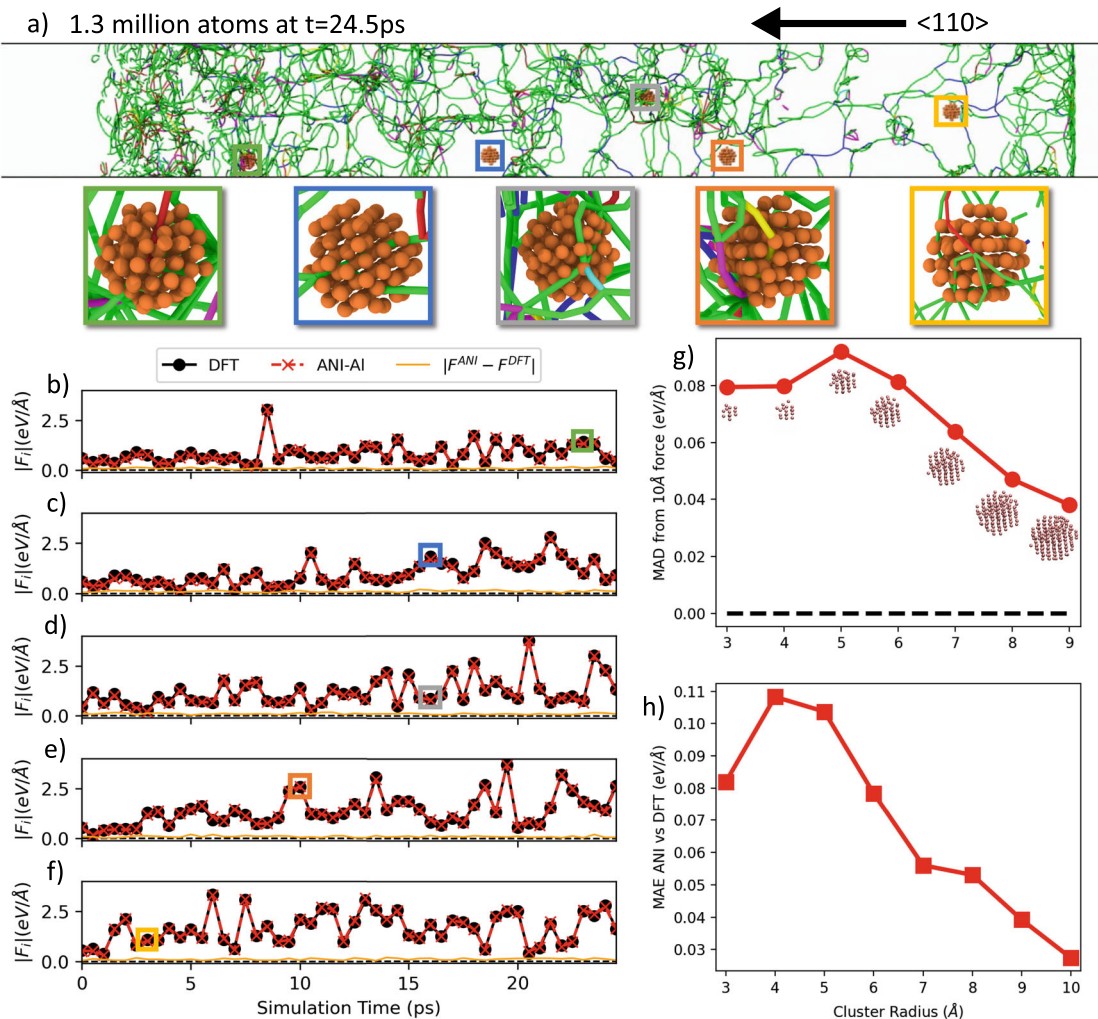

**Fig. 7 A 1.3 million atom shock simulation using the ANI-Al potential.** A shock of 1.5 km/s was initiated from the right along the ⟨110⟩ crystallographic direction. **a** Dislocation structure calculated using OVITO after 24.5 ps of simulation as well as zooms for five randomly selected atoms at hand-picked times. **b**–**f** Verification of the ANI-Al force predictions for these five atoms every 0.5 ps. Reference forces were obtained by performing new DFT calculations for small clusters centered the five atoms. **g** Comparison of DFT-calculated forces on the central atom for varying cluster radius (reference force calculated at radius 10 Å). **h** Mean absolute error of ANI-Al predicted forces, relative to DFT, as a function of cluster radius.

although the nominal ANI-Al cutoff radius is just 7 Å, the model can still generate strong effective couplings at distances of up to 10 Å through intermediary atoms that process angular features, as described in Supplementary Note 4.

**Limited vs. diverse sampling.** The success of ANI-Al hinges on the diversity of the active learned dataset. To demonstrate this, we compare ANI-Al against an ML model trained on a much more limited dataset. We will call this baseline dataset *FCC/Melt*, as it

Table 1 Mean absolute and root-mean squared errors (MAE/RMSE) of ANI-Al models trained/tested on combinations of FCC/Melt (FM) and active learning (AL) datasets.

|  | FM tested | AL tested |
|---|---|---|
| | *Energy error* (meV/atom) | |
| FM trained | 2.0/4.0 | 40/110 |
| AL trained | 1.4/1.9 | 1.3/1.9 |
| | *Force component error* (eV/Å) | |
| FM trained | 0.04/0.07 | 0.49/1.53 |
| AL trained | 0.03/0.04 | 0.04/0.06 |

consists only of samples from the FCC and liquid phases. Specifically, the FCC/Melt dataset is constructed by taking regular snapshots from near-equilibrium MD trajectories. For each snapshot, we perform a DFT calculation to determine the reference energy and forces.

The first such MD trajectory is shown in Fig. 6. There, 108 atoms were initialized to FCC, heated from 300 to 1500 K, and cooled back to 300 K. We take 300 snapshots from this trajectory, equally spaced in time, to add to the FCC/Melt dataset. For increased variety, the FCC/Melt dataset contains an additional 250 DFT calculations taken from the liquid phase over a range of temperatures and pressures (Supplementary Note 7 contains details). In sum, the FCC/Melt dataset contains 550 DFT calculations for near-equilibrium FCC and liquid configurations.

Table 1 compares our ANI-Al model, trained on the full active learned (AL) dataset, to an ANI model trained on the much more restricted FCC/Melt (FM) dataset. The two model types are compared by testing on held out portions of both datasets. Supplementary Figs. 5 and 6 show the associated correlation plots.

A conclusion of Table 1 is that both the AL trained and FCC/Melt-trained models to have comparable errors when predicting on the held out FCC/Melt test data. However, when testing on the held out AL data, the FCC/Melt-trained model does quite poorly. This failure casts doubt on the ability of the FCC/Melt-trained model to study new dynamical physical processes: will a rare event occur that pushes the FCC/Melt-trained model outside its range of validity? To mitigate this danger, it is essential to make the training dataset as broad as possible, which is our aim with active learning.

**Coverage of configuration space**. Here, we characterize the sampling space covered by our active learning methodology using the t-distributed stochastic neighbor embedding[81] (t-SNE) method as implemented in the OpenTSNE[82] Python package. In Fig. 8a–d, every local atomic environment in the active learned training dataset is mapped onto a reduced, two-dimensional space. Hyperparameters of the t-SNE-embedding process is shown in Supplementary Note 6. In brief, we use the activations after the first layer of the ANI-Al neural network as an abstract representation ("latent space vector") of the 7 Å-radius local atomic environments around each atom. The cosine distances between all pairs of these latent space vectors (for all points of the dataset) are the inputs to t-SNE. The output of t-SNE is, ideally, a mapping of all latent space vectors onto the two-dimensional embedding space that, in some sense, is maximally faithful to pairwise distances. t-SNE thus provides a two-dimensional visualization of all atoms in all configurations of the dataset.

Figure 8a–d uses radial neighbor regression (RNR) to associate atomic environments (averaged within the embedding space) with four different properties. Figure 8a shows the average active

learning iteration count, Fig. 8b shows the average force error (saturated at 0.5 eV/Å), Fig. 8c shows the ANI predicted atomic energy (saturated at 1.5 eV), and Fig. 8d shows the trace of the ANI-Al predicted atomic stress tensor (saturated at 0.025 eV/Å).

Observe that the sampled points are well connected in the reduced dimensional space, and not clustered. In contrast, a poorly sampled dataset would typically lead to obvious clusters, clearly separated by gaps. In Fig. 8a, one sees that the active learning procedure progresses from sampling random disordered configurations (blue region at the top) to sampling much more structured data. The left, bottom, and right edges of the embedding space were not sampled until late in the active learning process (red). Up until about ten iterations into the active learning procedure, all MD sampling trajectories never ran long enough to make it to an ordered atomic configuration (recall that the trajectories end once they reach a configuration with very high ML uncertainty). Despite being very well-sampled, a comparison with Fig. 8b shows that the ML model still has the greatest difficulty in fitting this disordered (high entropy) region of configuration space. Figure 8c, d shows that these disordered atomic environments typically have high energies and stresses.

Markers in Fig. 8b show the local atomic environments for perfect crystals; we selected eight crystal structures that could potentially compete with FCC as the ground state. Observe that all eight markers lie within the sampled space (interestingly, only FCC and A15 crystals are placed at the edge), and are continuously connected. The average force error in the region of all crystal structures is generally very low (less than 0.1 eV/Å), except for the simple cubic and diamond cubic regions, which are very high-energy structures, and thus less physical. Figure 8c shows that the position of FCC is almost perfectly overlapping the lowest energy configuration sampled during active learning. As mentioned above, the FCC structure was not found until at least 10 active learning iterations. Later in the active learning process, however, local FCC configurations became quite well-sampled (cf. Fig. 8a).

Red crosses in Fig. 8d represent local atomic environments that were randomly sampled from our previous shock simulation. Interestingly, these samples are largely confined to the bottom-right portion of embedding space and span a fairly significant range of local atomic stresses. Early in the shock simulation, the atomic environments live primarily near the FCC region of the embedding space, with small local stress. As the shock wave passes through each local environment, one can sample much higher pressure and temperature conditions. Afterward, there remains a complicated pattern of defects. Importantly, throughout the entire shock process, all visited atomic environments appear to be well represented by the training dataset. This is consistent with the fact that the force errors of Fig. 8b appear to remain small for all regions (e.g., bottom-edge of embedding space) where the shocked environments appear. The region circled and labeled "Liquid phase sampling" was obtained from the atomic environments in the liquid phase simulations shown in Supplementary Fig. 4 and described in Supplementary Note 7. The configurations appearing in a shock are largely distinct from those appearing in simulations of the liquid phase.

**Discussion**

ML is emerging as a powerful tool for producing interatomic potentials with unprecedented accuracy; recent models routinely achieve errors of just a couple meV per atom, as benchmarked over a wide variety of ordered and disordered atomic configurations. Here, we presented a technique to automatically construct general-purpose ML potentials that requires almost no expert knowledge.

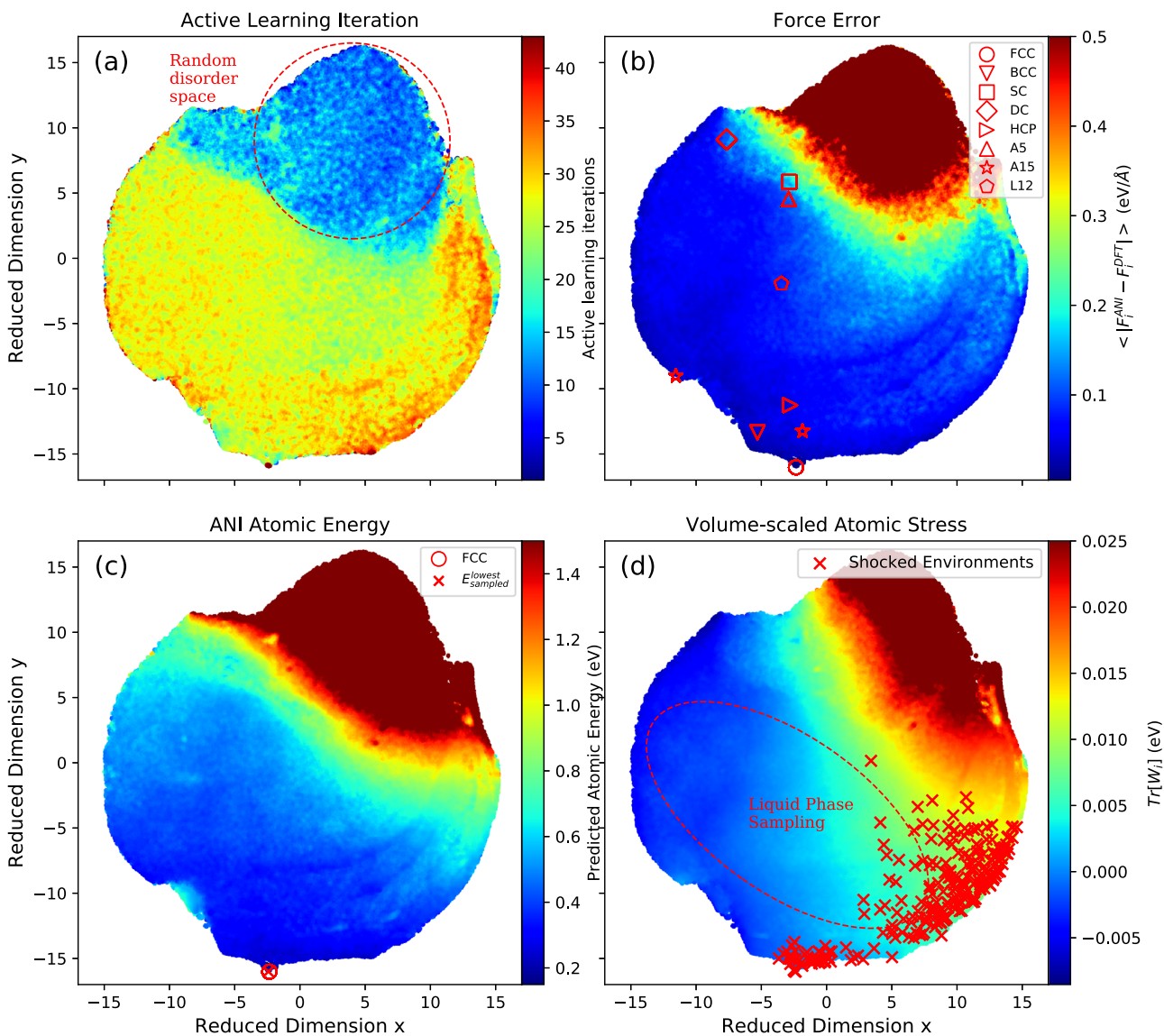

**Fig. 8 Visualization of the atomic configurations sampled by active learning.** We used the t-distributed stochastic neighbor embedding (t-SNE) method to map local atomic environments into two dimensions. Radial neighbor regression (RNR) is used to color the average property within a radius of a given point in the 2D embedding space. **a** Active learning iteration at which a sample was taken; disordered space is circled. **b** Force error; eight crystal structures are marked. **c** ANI predicted atomic energy; FCC is observed to be the lowest energy configuration in the embedding space. **d** Volume-scaled atomic stress; shocked environments are marked and liquid environments are circled.

Modern ML potentials can be used for large-scale MD simulations. To quantify performance, consider for example the optimized Neurochem code applied to ANI-Al with an 8× ensemble of neural networks, and a simulation volume of thousands of atoms; here, we measure up to 67 k atom timesteps per second when running on a single Nvidia V100 GPU. With 80 GPUs, our current LAMMPS interface (not fully optimized) achieved 1.6 M atom timesteps per second for the 1.3 M-atom shock simulation. A study conducted parallel to ours performed ML-MD simulations of 113 M copper atoms by using 43% of the Summit supercomputer (~27 k V100 GPUs)[83]. The speed of ANI-Al is perhaps two orders of magnitude slower than an optimized EAM implementation, but vastly faster than ab initio MD would be.

Because ML models are so flexible, the quality and diversity of the training dataset is crucial to model accuracy. Here, we focused on the task of dataset construction and, specifically, sought to push the limits of active learning. We presented an automated procedure for building ML potentials. The required inputs

include physical parameters such as the temperature and density ranges over which to sample, the interaction cutoff radius for the potential (we selected 7 Å for aluminum), and various ML hyperparameters that we reused from previous studies. We did not include any expert knowledge about candidate crystal ground states, defect structures, etc. Nonetheless, the active learning procedure eventually collected sufficient data to produce a broadly accurate potential for aluminum.

We emphasize that the starting point for the active learning procedure consisted of DFT calculations for completely disordered configurations. As the ML potential improved, the quality of the MD sampling increased, and the training data collected could become more physically relevant. The timeline of this process is illustrated in Fig. 8a. After about ten active learning iterations (1000+ DFT calculations), the ML potential became robust enough that the MD simulations could nucleate crystal structures. From this point onward, the ML predictions for crystal properties could rapidly improve.

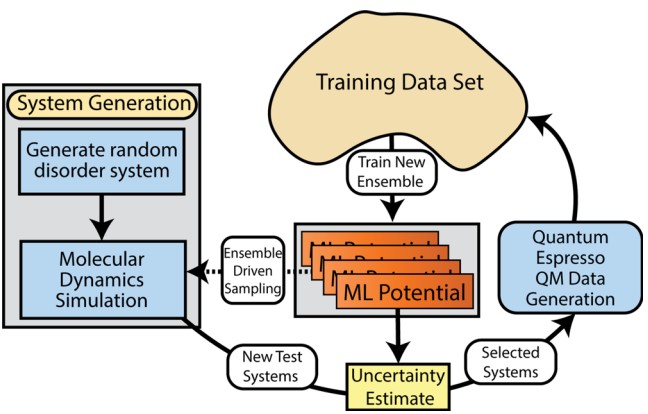

**Fig. 9 Diagram of active learning for data collection.** Multiple active learning cycles can be run simultaneously, with occasional synchronization points to merge new data into a single global dataset. The sampling, data generation, and training steps all benefit from graphics processing unit (GPU) acceleration.

Previous potential development efforts have benefited from careful dataset design. Our decision to pursue a fully automated approach certainly made the modeling task more difficult but was motivated by our belief that defects appearing in real, highly nonequilibrium processes may be difficult to characterize a priori. As an example, consider the complicated dislocation patterns appearing in the shock simulation of Fig. 7. It would likely be difficult to hand-design a dataset that fully captures all defect patterns appearing in shock. Active learning, however, seems to do a good job of sampling the relevant configuration space (cf. the marked points in Fig. 8d). Indeed, throughout the entire shock simulation, the ANI-Al predicted forces are in good agreement with new reference DFT calculations, even for atoms very near dislocation cores. Even though most of the active learned training data is far from perfect FCC, the ability of ANI-Al to predict aluminum FCC properties seems roughly in line with other recent ML studies, as shown in Supplementary Table 4[32,34].

A challenge for the active learning procedure presented in this work is its large demand on computational resources. Our final active learned dataset contains over 6000 DFT calculations; each calculation was performed on a supercell containing up to 250 atoms. For future work, it would be interesting to explore whether the majority of the training data could be weighted toward much smaller supercells. It would also be interesting to investigate ways to make active learning more efficient, e.g., by systematically studying the effect of various parameters required by the procedure. Other areas for improvement may include: employing a dynamics with modulated stress or strain, smarter sampling that goes beyond nonequilibrium MD[84], and better estimation of the ML error bars.

## Methods
This section presents details of the automated procedure to build ANI-Al, our general-purpose machine-learning potential for bulk aluminum.

**The ANI machine-learning model**. ANI is a neural network architecture for modeling interatomic potentials. Our prior work with ANI has largely focused on modeling clusters of organic molecules[25]. A variety of ANI potentials are available online (cf. "Code availability"). Here, we presented ANI-Al, our ANI model for aluminum in both crystal and melt phases.

Our training data consist of DFT calculations, evaluated on "interesting" atomic configurations, as identified by an active learning procedure. We used the PBE functional, with parameters described in Supplementary Note 1. One point to mention is that our $3 \times 3 \times 3k$-space mesh was, in retrospect, perhaps too small. For the varying box sizes of our training data, this corresponds to 31.5–51 $k$-points per Å$^{-1}$. A more careful choice would be 57 $k$-points per Å$^{-1}$ independent of system size[85].

The input to ANI is an atomic configuration (nuclei positions and species). To describe these configurations in a rotation and translation invariant way, ANI employs Behler and Parrinello[5] type atomic descriptors, but with modified angular symmetry functions[20]. Details of all model hyperparameters are provided in Supplementary Note 3. The most important hyperparameter is the 7 Å interaction cutoff distance, which we selected based on careful trial and error. Other hyperparameters, such as the number of symmetry functions, were largely reused from the previous studies[25]. ANI's total energy prediction is computed as a sum over local contributions, evaluated independently at each atom. Each local energy contribution is calculated using knowledge of all atoms within the 7 Å cutoff. Using backpropagation, one can efficiently calculate all forces as gradients of the predicted energy.

Each DFT calculation outputs the total system energy $E$ and the forces $\mathbf{f}_j = \partial E/\partial \mathbf{r}_j$ for all atoms $j = 1 \ldots N$. Our loss function for a single DFT calculation,

$$L \propto \left(\hat{E} - E\right)^2 + \ell_0^2 \sum_{j=1}^{N} \left(\hat{\mathbf{f}}_j - \mathbf{f}_j\right)^2, \tag{1}$$

is a measure of disagreement between the ANI predictions for energy, $\hat{E}$, and forces, $\hat{\mathbf{f}}_j = \partial \hat{E}/\partial \mathbf{r}_j$, and the DFT reference data. A length hyperparameter $\ell_0$ is empirically selected so that energy and force terms have comparable magnitude. In our tests, the specific choice of $\ell_0$ did not strongly affect the quality of the final model.

Training ANI corresponds to tuning all model parameters to minimize the above loss, summed over all DFT calculations in the dataset. For stochastic gradient descent, each training iteration requires estimating the $\partial L/\partial W_i$ for all model parameters $W_i$ (in our case, there are order $10^5$ parameters). Because forces $\hat{\mathbf{f}}_j$ appear in $L$, calculating $\partial L/\partial W_i$ seems to involve all *second* derivatives of the ANI energy output, i.e., $\partial^2 \hat{E}/\partial W_i \partial \mathbf{r}_j$. Fortunately, direct calculation of these can be avoided. Instead, we employ the recently proposed method of Ref. 86 to efficiently calculate all $\partial L/\partial W_i$ in the context of our C++ Neurochem implementation of ANI. A brief summary of the method is presented in Supplementary Note 5. With this method, the total cost to calculate all $\partial L/\partial W_i$ is within a factor of two of the cost to calculate all forces.

To improve the quality of our predictions, the angle ANI-Al model actually employs ensemble-averaging over eight neural networks. Each neural network in the ensemble is trained to the same data but using an independent random initialization of the model parameters. We observe that ensemble-averaged energy and force errors can be up to 20% and 40% smaller, respectively, than those of a single neural network prediction.

**Active learning overview**. The active learning process employed here is similar to that in previous work[41], adapted for materials problems and efficient parallel execution on hundreds to thousands of nodes on the Sierra supercomputer. We first train an initial model to a dataset of about 400 random disordered atomic configurations. Next, we begin the AL procedure, as illustrated in Fig. 9. Using the current ML potential, we simulate many MD trajectories, each initialized to a random disordered configuration. During these simulations, the temperature is dynamically varied to diversify the sampled configurations. As these MD simulations run, we look at the variance of the predictions for the eight neural networks within an ensemble to determine whether the model is operating as expected[87]. Prior work indicates that this measured ensemble variance correlates reasonably well with actual model error[41]. If the ensemble variance exceeds a threshold value, then it seems likely that collecting more data would be useful to the model. In this case, MD trajectory is terminated and the final atomic configuration is placed on a queue for DFT calculation and addition to the training dataset. Periodically, the ML model is retrained to the updated training dataset. This AL loop is iterated until the cost of the MD simulations becomes prohibitively expensive. Specifically, we terminate the procedure when typical MD trajectories reach about 250a ps (about $2.5 \times 10^5$ timesteps) without uncovering any weaknesses in the ML model. The final active learned dataset contains 6352 DFT calculations, each containing 55–250 atoms, and having varying levels of disorder.

We emphasize that this active learning procedure is fully automated, and receives no direct guidance regarding atomic configurations of likely relevance, such as crystal structures. The initial training dataset consists only of disordered atomic configurations, and every MD simulation is initialized to a disordered configuration. The MD simulations use only forces as predicted by the most recently trained ML potential. After many active learning iterations, the MD simulations will hopefully be sufficiently robust to support nucleation into, e.g., the crystal ground state, and then the active learning scheme can begin to collect this type of training data. In this sense, the active learning scheme must automatically discover the important low energy and nonequilibrium physics.

Supplementary Note 2 gives further details regarding the active learning procedure.

**Randomized atomic configurations**. We employ randomized atomic configurations to collect an initial dataset of DFT calculations and to initialize all MD simulations for AL sampling. The procedure to randomize a supercell is as follows:

1. Randomly sample each of the three linear dimensions of the orthorhombic supercell uniformly from the range 10.5–17.0 Å.
2. Randomly select a target atomic density $\rho$ uniformly from the range 1.80–4.05 g/cm$^3$.
3. Iteratively place atoms randomly in the supercell. If the proposed new atom lies within a distance $r_{\min} = 1.8$ Å of an existing atom (i.e., roughly the van der Waals radius), that placement is rejected as unphysical. The placement of atoms is repeated until the target density $\rho$ has been reached.

**Nonequilibrium temperature schedule**. To maximize the diversity in active learning sampling, we perform the MD simulations with a Langevin thermostat using a temperature that varies in time according to a randomized schedule. Compared with previous work that sampled from a specific temperature quench schedule[88], here we employ a more diverse and randomly generated collection of temperature schedules.

Starting at time $t = 0$, and running until $t = t_{\max} = 250$ fs, the applied temperature is,

$$T(t) = T_{\text{start}} + \frac{t}{t_{\max}}(T_{\text{end}} - T_{\text{start}}) + T_{\text{mod}}\sin^2(\pi t/t_0) \quad (2)$$

The first two terms linearly ramp the background temperature. The initial temperature $T_{\text{start}}$ is randomly sampled from the range 10–1000 K. The final background temperature $T_{\text{end}}$ is randomly sampled from the range 10–600 K. The third term in Eq. (2) superimposes temperature oscillations. The modulation scale $T_{\text{mod}}$ is randomly sampled from the range 0–2000 K. The oscillation period $t_0$ is randomly sampled from the range 10–50 ps.

By spawning MD simulations with many different temperature schedules, we hope to observe a wide variety of nonequilibrium processes. Given that each MD simulation begins from a disordered melt configuration, we hope that the nonequilibrium dynamics will automatically produce: (1) nucleation into various crystal structures (in particular, the ground-state FCC crystal), (2) a variety of defect structures and dynamics (dislocation glide, vacancy diffusion, etc.) and (3) rapid quenches into disordered glass phases. Acquiring snapshots from these types of dynamics will be crucial to the diversity of the training dataset and, thus, to the overall generality of the ANI-Al potential.

## Data availability
The active learned training dataset and final ANI-Al potential are available at https://github.com/atomistic-ml/ani-al.

## Code availability
Two implementations of the ANI neural network architecture are available online: TorchANI (https://github.com/aiqm/torchani) and NeuroChem (https://github.com/atomistic-ml/neurochem).

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

## Acknowledgements

This work was partially supported by the LANL Laboratory Directed Research and Development (LDRD) and the Advanced Simulation and Computing Program (ASC) programs. Within ASC, we acknowledge support from the Physics and Engineering Modeling (ASC-PEM) subprogram and the Advanced Technology Development and Mitigation (ASC-ATDM) subprogram. We acknowledge computer time on the Sierra supercomputing cluster at LLNL, Institutional Computing at LANL, and the CCS-7 Darwin cluster at LANL. J.S.S. was supported by the Nicholas C. Metropolis Postdoctoral Fellowship. N.M. and J.S.S. were partially supported by the Center for Nonlinear Studies (CNLS). This work was performed, in part, at the Center for Integrated Nanotechnologies, an Office of Science User Facility operated for the U.S. Department of Energy (DOE) Office of Science.

## Author contributions

J.S.S., B.N., and N.L. designed, implemented, and applied the ML methodology. N.M., J.C., L.B., and S.F. performed and analyzed the large-scale MD simulations, and calculated materials properties for classical and ML potentials, as well as DFT. H.A.M. adapted codes for execution on the Sierra supercomputer. J.S.S., B.N., S.T., T.G., S.F., and K.B. planned the study and wrote the paper.

## Competing interests

The authors declare no competing interests.

**Additional information**

