## [Peer Review File · Nature Communications]

Reviewers' comments:

Reviewer #1 (Remarks to the Author):

In this manuscript the authors detail their work on constructing a machine learned interatomic potential for aluminum. They use a feed-forward artificial neural network and Behler-Parrinello type atom centered symmetry functions (slightly modified) as descriptors (the first author has extensive experience with this machinery because he was also part of the team that made ANI, the first serious machine learning based force field for organic molecules). I find the manuscript well written, between them the authors are clearly experts in both modelling metals and the application of machine learning to materials modelling.

In assessing the potential impact and importance of the work, I am a bit struggling to articulate the main point of the manuscript, the "take-home message", the reason why a fellow researcher will want to note the results and build on this work. There are several directions in which this could happen.

(i) The paper is providing a potential for a metal that is often used in materials modelling as the "simplest system" to try things out on, and it even has a scientific interest albeit at very high pressures (AI is used as the pressure benchmark in ultra-high pressure experiments). However, there are not enough tests to demonstrate that this potential itself will actually be useful as it is for materials modellers who are themselves not in the business of making potentials. It is clearly a good start, and there a few things the authors could do to provide further benchmarks, see below in the detailed list of comments.

(ii) The paper is showing *how* to make a potential for materials using active learning and molecular dynamics. In this respect, the authors cite numerous previous works that have demonstrated active learning approaches to select the database for fitting. They adapt these to their chosen methodology. The novelty in the manuscript is the use of cycling pressure/temperature in the molecular dynamics. (MD itself has been used before). If this is the main point of the paper, then I would expect the authors to go into more detail in analysing the consequences of the parameter settings of their scheme, to demonstrate the advantages of their scheme over more a more naive approach (e.g. just plain MD). For example, it is clear that the authors did some work to enable the NN scheme to fit to forces - but they are actively choosing NOT to talk about it and reserve that to a separate paper (and in any case, others e.g. Jorg Behler, have been fitting NN potentials to forces for nearly a decade - how does the authors' method compare, why is it necessary to change this etc?).

So as it stands, the manuscript forms the core of a nice piece of work, but in order to make it scholarly and impactful to the level of Nature Communications, the authors need to decide which direction to take it, and complete additional work and add further results and discussion. Alternatively, if left as it is, the paper can be published in a significantly less impactful journal, and its take home message is "We're on the way to making useful potentials using active learning, come back and have a look how we're doing in a year's time" - Scientific Reports, anyone? Below I include a long list of questions, comments and suggestions that will help the authors no matter which route they chose.

Detailed points (not in any particular order):

1. The authors do not make use of stresses, which are also given (essentially for free) by the DFT calculation. Elastic constants, acoustic phonons, etc will be better described, and presumably the same finite difference scheme for training should work equally well ?

2. The authors advocate and use active learning to sample a wide range of neighbour configurations. The ultimate test of this is random structure search (RSS), where by small and moderate sized unit cells (1-16 atoms) are initialised randomly and relaxed. This can easily be done with DFT, and obtaining comparable “density of states” with the potential will go a long way to convince readers that the potential is indeed robust.

3. The active learning procedure to generate samples has medium complexity. Which features of this procedure are essential? Presumably the authors increased this complexity step by step until they had a good scheme. What happened when they attempted to use simpler schemes (e.g. no temperature cycling, no pressure cycling, no wide range of temperatures etc) ? Readers would learn a lot from seeing at least some of the benchmarks from these less-than-adequate runs. This sort of information is what takes the field forward!

4. The authors say that some basic crystal structures are correctly reproduced even though the potential is not trained to them. This is not quite true, the AL procedure could have easily sampled the local environments of these crystal structures. This is not a problem of course, this is how AL works. But it would be interesting to see just which local environments were actually sampled, and how this impacts the accuracy of the energy-volume curves. The authors should produce a dimensionally reduced chart of their local environments (e.g. using PCA with the descriptors as dimensions, this is very simple to do), and mark interesting crystal structures and all their training data. It would be further interesting to see the evolution of this chart during the AL procedure, together with the energy-volume curves of the various high symmetry crystal phases from different points in the AL iterations.

5. Why is the bulk modulus compared to experiment rather (or as well as) to the DFT value? Rather than just in the text, please provide a table of all three elastic constants, showing experiment, DFT and ANI.

6. Please estimate the k-point convergence error of DFT numerically. The authors say that their k-point sampling was “too small”, this strikes me as poor scholarship. If they know it’s too small, then they should redo their work (it only takes 5 days!) with better k-point sampling. I recommend fixing the k-point density, rather than the grid. How does their grid compare to typical k-point densities that others have used for fitting ML models (e.g. 0.2 in units that VASP uses).

7. The authors comment on the description of the liquid, suggesting that finite electronic temperature is the reason for the discrepancy of their RDFs. I think this excuse doesn’t stack up. I am pretty sure that they used finite electronic smearing in their calculations, which corresponded to a much higher temperature than where their tests were run. Many papers have fitted metallic liquids before using such DFT data, and RDFs should match extremely well. Is it possible that the authors mistakenly used the zero temperature extrapolated energies in the training, rather than the free energy reported by the DFT code? This would have been a silly thing to do, because it is the free energy rather than the extrapolated zero temperature energy that is the quantity whose derivatives are given by the Hellman-Feynman forces, and so the data would be inconsistent.

8. Please also show angular distributions for the liquid and the equation of state (density as a function of temperature) with your chosen DFT.

9. Please show the ANI prediction for the 0K transition pressures. I am sure DFT has been done up to very high pressures, and looking at those extrapolation would be interesting (even if ANI fails, we need to know where! This is scholarship, rather than a beauty parade)

10. Please show the melting line all the way up to the BCC transition. This is the sort of calculation that a potential is actually useful for, because DFT is so expensive!

11. Any proposed potential that is to be used by modellers needs to do well for defects. Even demonstrating machine learning protocols need to do that: they need to demonstrate that they can cope with defects. The authors our scorn on previous potential making protocols because they were designed specifically to handle defects. Automating the database generation is great, and that's part of the story in the paper, but of course the authors are then obliged to show that their automated procedure doesn't need hand-designed defect data, but still gets defects right! The authors present the defect results in the supplementary, and frankly they are not that great. Interstitial formation energy is way off, the elastic constants should be a lot better. This needs to be put into the main part of the manuscript (not the entire table just the ANI-DFT-EXP comparison), because this goes to the heart of what the authors are trying to say. Precisely because real world simulations do care about defects, as the authors are eager to point out themselves.

12. The authors could do well to add more complex defects, e.g. the pair of edge dislocation partials (studied to death by explicit DFT before). One of biggest omissions of the current scheme the authors propose is the lack of surfaces (as far as I could tell). It is nevertheless interesting that surface energies are not bad. This again should be in the main text. How would the training protocol look like if one wanted to include surfaces in the training data?

13. The authors formulate their main message as "In the present study, we demonstrate that it is possible to obtain a broadly accurate potential for aluminum with essentially zero human guidance. " What about hand tuning of descriptor parameters? How would one do that for a new system? Especially one with multiple elements?

14. At least the final potential needs to be available for others to test and verify the paper. The DFT training data needs to be made available so that others can scrutinise the potential training, and build on the current work in terms benchmarking.

15. The authors use a committee scheme for predicting the error of their model and to drive the active learning protocol. Is there evidence for correlation of committee disagreement with actually error? it's not clear that this is actually necessary for the protocol to work, but would be very interesting to look at. It is entirely possible that the committee drive exploration, regardless of any correlation with the error.

16. What happens if tolerance to include new configurations is loosened or tightened? Is the resulting model final more/less accurate? Again showing this data is very helpful and part of scholarly work on active learning.

I hope the authors are willing to take on board my points and develop their work accordingly.

Reviewer #2 (Remarks to the Author):

The manuscript describes the development and validation of a machine-learning potential for bulk aluminium, using an active learning scheme that requires little human interaction. The resulting potential is validated on bulk properties of aluminium as well as phase transition simulations.

The novelty of the approach compared to previous work is the level of automation, although compared to e.g. 10.1103/PhysRevB.95.094203 it is really very incremental. Given that one of the main claims

is the correctness of the bulk phases, I think the validation is incomplete: phonon spectra are missing, for example. I also think that in e.g. Fig. 4 the authors should have compared to DFT results rather than experiments - which indeed should be included in the figure, but the target is DFT, so the validation should be against DFT. This is methodological work, after all. Finally, I didn't find the argument that FCC is the most stable structure compelling, I suggest using a more thorough minima-searching algorithm. What's completely missing from the paper are important and interesting defects, e.g. surfaces, interstitials etc.

Overall, I think my questions should be addressed to prove that this is a usable interatomic potential at all, and even then I am not sure this protocol represents a large enough advance of the field to warrant wider interest than that of the materials-ML community.

Reviewer #3 (Remarks to the Author):

The authors present a neural network potential for Al developed using an active learning (AL) approach. It is demonstrated that the potential has some advantages over the traditional potentials for certain properties that were tested. The potential is claimed to be general-purpose type and "discovered automatically".

The following critical comments/questions should be addressed before I can continue evaluation of this paper.

1. The fact that the potential reproduces the ground-state FCC structure and a few other crystal structures is presented as a "discovery" of these structures, which I think is an exaggeration. The training database contained information about the potential energy surface of Al in relevant regions, and in particular included configurations containing crystals nucleated from the melt. MD was run at temperatures from 1000 K to 10 K to explore the energy landscape, so the training data set was fully aware of the low-energy configurations. I do not see how this is surprising.

2. The AL approach is widely used almost everywhere these days, including the construction of ML potentials. Examples are given by the previous paper of these authors [45], by Ref.34, and other recent publications. What are the novel aspects of the proposed AL approach? Or is this a mere extension/application of Ref.45 and other recent papers to this particular metal?

3. It is claimed the proposed potential is general-purpose type and outperforms the traditional potentials. To substantiate this strong claim, the authors should test their potential for a wider set of properties for which the traditional potentials are routinely tested. As a minimum, this set of properties should include:

3A. Comparison with a larger number of alternate crystal structures. In particular, the A15 structure should be shown, which often competes for the ground state with FCC.

3B. To demonstrate transferability, the energy-volume functions (Fig.2) should be extended to volumes beyond those covered by the DFT calculations, including stronger compressions and larger expansions, all the way to the cutoff volume.

3C. Other deformation paths, for example the $\langle 111 \rangle$ tension and compression.

3D. Potential predictions (and comparison with traditional potentials) for:

- phonon dispersion curves
- thermal expansion coefficient as a function of temperature
- surface energies and surface relaxations for different crystallographic indices
- point-defect formation and migration energies (including different interstitial orientations)

Demonstration of superior performance of the potential for just a few selected properties is not sufficient for claiming transferability to almost all other properties (which is what "general-purpose" means). Otherwise the practical usefulness of the proposed potential can be questionable. If there is no substantially superior performance across a wide spectrum of properties, then a traditional potential might as well be utilized (which is way faster).

4. Comparison with other ML potentials for Al should be made, including, for example, J. Chem. Phys. 148, 241733 (2018) and Nat. Comm. 10, 2339 (2019).

5. The acronym ANI has not been defined in this paper.

Reviewer #1 (Remarks to the Author):

Referee Summary:

machine learned interatomic potential for aluminum. They use a feed-forward artificial

neural network and Behler-Parrinello type atom centered symmetry functions (slightly modified) as descriptors (the first author has extensive experience with this machinery because he was also part of the team that made ANI, the first serious machine learning based force field for organic molecules). I find the manuscript well written, between them the authors are clearly experts in both modelling metals and the application of machine learning to materials modelling.

Comment: *In assessing the potential impact and importance of the work, I am a bit struggling to articulate the main point of the manuscript, the “take-home message”, the reason why a fellow researcher will want to note the results and build on this work. There are several directions in which this could happen.*

Response: We thank the reviewer for raising this important concern, and have addressed it through significant revisions to the manuscript text, and inclusions of new results (e.g. a DFT-verified shock simulation). More details are described at the beginning of this response.

Comment: *(i) The paper is providing a potential for a metal that is often used in materials modelling as the “simplest system” to try things out on, and it even has a scientific interest albeit at very high pressures (Al is used as the pressure benchmark in ultra-high pressure experiments). However, there are not enough tests to demonstrate that this potential itself will actually be useful as it is for materials modellers who are themselves not in the business of making potentials. It is clearly a good start, and there are a few things the authors could do to provide further benchmarks, see below in the detailed list of comments.*

Response: This point was also raised by other Reviewers, and is a good one. The revised manuscript includes nearly all requested new materials properties measurements. These include point defects, surface measurements, phonon spectrum, trigonal deformation measurements, etc. Many more property benchmarks, and comparisons with previous potentials, are now shown in Tables S3 and S4 of the supplemental information (SI). The new Figures 8 and 9 of the main text (shown a shock simulation, and a visualization of the diversity of the sampled space) are very significant additions that illustrate what we consider to be important advances.

Comment: *(ii) The paper is showing *how* to make a potential for materials using active learning and molecular dynamics. In this respect, the authors cite numerous previous works that have demonstrated active learning approaches to select the database for fitting. They adapt these to their chosen methodology. The novelty in the*

manuscript is the use of cycling pressure/temperature in the molecular dynamics. (MD itself has been used before). If this is the main point of the paper, then I would expect the authors to go into more detail in analysing the consequences of the parameter settings of their scheme, to demonstrate the advantages of their scheme over more a more naive approach (e.g. just plain MD). For example, it is clear that the authors did some work to enable the NN scheme to fit to forces - but they are actively choosing NOT to talk about it and reserve that to a separate paper (and in any case, others e.g. Jorg Behler, have been fitting NN potentials to forces for nearly a decade - how does the authors' method compare, why is it necessary to change this etc?).

Response: We thank the reviewer for these suggestions.

If it were feasible, we wish we could do a more systematic study of the consequences of various parameter settings in the active learning scheme. A serious practical problem for doing so is computational expense, which we now explicitly state in the Outlook section:

A challenge for the active learning procedure presented in this work is its large demand on computational resources. Our final active learned dataset contains over 6,000 DFT calculations; each calculation was performed on a supercell containing up to 250 atoms. For future work, it would be interesting to explore whether the majority of the training data could be weighted toward smaller supercells. It would also be interesting to investigate ways to make the active learning more efficient, e.g., by systematically studying the effect of various parameters required by the procedure.[...]

Training to force data is a technical topic, now discussed in full detail in a separate manuscript: <https://arxiv.org/abs/2006.0547> . We have revised the present manuscript to cite this recent work. We have also added Sec. 1.5 of the SI to summarize the final equations needed for force training. The advantage of the new scheme is now stated in the text as follows:

With this method, the total cost to calculate all $\partial L / \partial W_i$ [i.e., the full loss gradient] is within a factor of two of the cost to calculate all forces.

Comment: So as it stands, the manuscript forms the core of a nice piece of work, but in order to make it scholarly and impactful to the level of Nature Communications, the authors need to decide which direction to take it, and complete additional work and add

further results and discussion. Alternatively, if left as it is, the paper can be published in a significantly less impactful journal, and its take home message is “We’re on the way to making useful potentials using active learning, come back and have a look how we’re doing in a year’s time” - Scientific Reports, anyone? Below I include a long list of questions, comments and suggestions that will help the authors no matter which route they chose.

Response: With this resubmission, we have substantially expanded the scope of the manuscript. Below we address in detail all the Reviewer’s helpful comments/suggestions. Besides adding many materials property benchmarks, we also include the new Fig. 8 (a verified shock simulation of 1.3M atoms) and Fig. 9 (a t-SNE analysis of configuration space sampled by active learning). We have also expanded the final Outlook section to discuss these new results. The revised manuscript has significantly benefited from Reviewer feedback, and we hope the Reviewers will agree with our assessment that it now rises to the level of Nature Communications.

Detailed points (not in any particular order):

Comment: 1. *The authors do not make use of stresses, which are also given (essentially for free) by the DFT calculation. Elastic constants, acoustic phonons, etc will be better described, and presumably the same finite difference scheme for training should work equally well?*

Response: We thank the reviewer for this suggestion. First we’ll make a comment, which may perhaps be obvious: Although we do not explicitly train to stresses, we do train to forces, which uniquely determine stress through the atomic virial. Figure 7 of the main text and Fig. S3 of the SI show performance in ANI-AI stress predictions. We have modified the following text in Sec. III-I,

Interestingly, there is a systematic tendency for ANI-AI to overestimate pressure, especially at negative pressures. This seems a bit surprising, because the ANI-AI force predictions seem reasonably good, and these determine pressure through the atomic virial tensor. Perhaps the tendency to ANI-AI overestimate pressure is a reflection the fact that a large fraction of its training data was sampled at very large positive pressures (cf. Fig. S1 in the SI)

It is an interesting idea to add an explicit stress term to the loss function which could be used to help correct these systematic errors, but we did not explore that in the paper.

The revised manuscript now shows ANI-AI predicted phonon spectra (Fig. 5) which seems to be in reasonable agreement with DFT-calculations.

Comment: 2. *The authors advocate and use active learning to sample a wide range of neighbour configurations. The ultimate test of this is random structure search (RSS), where by small and moderate sized unit cells (1-16 atoms) are initialised randomly and relaxed. This can easily be done with DFT, and obtaining comparable “density of states” with the potential will go a long way to convince readers that the potential is indeed robust.*

Response: The suggestion of random structure search (RSS) is an excellent idea that we will keep in mind for future studies.

To help illustrate the global generality of our potential, we now include Fig. 9, which visualizes the overall configurational space sampled by our automated active learning procedure (inspired by the Reviewer suggestion below). Figure 9b shows that our model is highly accurate for force prediction across the subset of configurational space that represents physically relevant configurations, e.g. liquid under multiple conditions, nine crystal types, shock dislocations, etc.

Comment: 3. *The active learning procedure to generate samples has medium complexity. Which features of this procedures are essential? Presumably the authors increased this complexity step by step until they had a good scheme. What happened when they attempted to use simpler schemes (e.g. no temperature cycling, no pressure cycling, no wide range of temperatures etc) ? Readers would learn a lot from seeing at least some of the benchmarks from these less-than-adequate runs. This sort of information is what takes the field forward!*

Response: We fully agree with the value of the proposed discussion. It is true that we initially tried several smaller scale experiments before running active learning at large scale; however, at the time, there were many simultaneously changing variables and it would be a struggle to summarize the lessons learned in a coherent way.

It is a great idea for a future study to rerun a (smaller scale) active learning study with systematically varied parameters. We now make suggestions along these lines in the final paragraph:

For future work, it would be interesting to explore whether the majority of the training data could be weighted toward smaller supercells. It would also be

interesting to investigate ways to make the active learning more efficient, e.g., by systematically studying the effect of various parameters required by the procedure. Other areas for improvement may include: employing a dynamics with modulated stress or strain, smarter sampling that goes beyond nonequilibrium MD, and better estimation of the ML error bars.

Comment: 4. *The authors say that some basic crystal structures are correctly reproduced even though the potential is not trained to them. This is not quite true, the AL procedure could have easily sampled the local environments of these crystal structures. This is not a problem of course, this is how AL works. But it would be interesting to see just which local environments were actually sampled, and how this impacts the accuracy of the energy-volume curves. The authors should produce a dimensionally reduced chart of their local environments (e.g. using PCA with the descriptors as dimensions, this is very simple to do), and mark interesting crystal structures and all their training data. It would be further interesting to see the evolution of this chart during the AL procedure, together with the energy-volume curves of the various high symmetry crystal phases from different points in the AL iterations.*

Response: Regarding the first point, we agree that active learning is successful if it can generate a training dataset that, in some sense, spans the range of atomic configurations of physical relevance. We revised the introduction to emphasize this point:

Crucially, the AL scheme receives no *a priori* guidance about the relevant configuration space it should sample. Nonetheless, after enough iterations [of MD+ML sampling], the AL procedure eventually encounters configurations that locally capture characteristics of crystals such FCC, HCP, and BCC.

The revised outlook now states:

As the ML potential improved, the quality of the MD sampling increased, and the training data collected could become more physically relevant. The timeline of this "discovery" process is illustrated in Fig.~\ref{fig:TSNE}a. After about 10 active learning iterations (1000+ DFT calculations), the ML potential became robust enough that the MD simulations could nucleate crystal structures.

The above excerpts also clarify that all MD sampling is driven by the most recent ML potential only (we make no use of *ab initio* MD, which would be significantly more expensive).

The previous version of the manuscript made a misleading statement which we have removed:

We emphasize that ANI-AI is not explicitly trained to any of these crystal structures.

The new manuscript instead makes statements like the following:

We emphasize that this active learning procedure is fully automated, and receives no direct guidance regarding atomic configurations of likely relevance, such as crystal structures.

We thank the Reviewer for their excellent suggestion to visualize the sampled configuration space via a reduced dimensional embedding. Our new Fig. 9 plots four different local properties in a 2D t-SNE embedding. Figure 9a evaluates the number of active learning iterations that were required to sample a given region of space. Figure 9b supplies the position of atomic environments for nine different crystal structures within this embedding space, overlaid on the the ANI-AI force error. The Reviewer's final suggestion ("evolution [...] of the energy-volume curves of the various high symmetry crystal phases from different points in the AL iterations") would be fascinating, but we could not include it due to time constraints.

Comment: 5. *Why is the bulk modulus compared to experiment rather (or as well as) to the DFT value? Rather than just in the text, please provide a table of all three elastic constants, showing experiment, DFT and ANI.*

Response: We added DFT elastic constants into the text at the beginning of Sec. IIIB. Several additional properties are shown in Table S3 of the SI, and compare EAM models, ANI-AI predictions, DFT reference calculations, and experiment. Table S4 of the SI compares against prior ML studies.

Comment: 6. *Please estimate the k-point convergence error of DFT numerically. The authors say that their k-point sampling was "too small", this strikes me as poor scholarship. If they know it's too small, then they should redo their work (it only takes 5 days!) with better k-point sampling. I recommend fixing the k-point density, rather than the grid. How does their grid compare to typical k-point densities that others have used for fitting ML models (e.g. 0.2 in units that VASP uses).*

Response: We thank the Reviewer for this comment. Section IIA of the revised manuscript now states:

One point to mention is that our 3x3x3 k-space mesh was, in retrospect, perhaps too small. For the varying box sizes of our training data, this corresponds to 31.5 to 51 k-points per inverse angstrom. A more careful choice would be 57 k-points per inverse angstrom [<https://onlinelibrary.wiley.com/doi/abs/10.1002/qua.24836>],

where the citation refers to a previous ML study of aluminum. As another point of comparison, the recent ML study, [<https://doi.org/10.1103/PhysRevMaterials.3.023804>], used a resolution of “0.08 Å⁻¹”, which we interpret to mean 12.5 k-points per inverse angstrom. Our k-space resolution appears to be somewhere in the middle of those two prior works.

After some testing, we did not find evidence that the k-space resolution was a significant source of error for the elastic constant measurements, so we removed that speculation from Sec. IIIC of the main text.

Comment: 7. *The authors comment on the description of the liquid, suggesting that finite electronic temperature is the reason for the discrepancy of their RDFs. I think this excuse doesn't stack up. I am pretty sure that they used finite electronic smearing in their calculations, which corresponded to a much higher temperature than where their tests were run. Many papers have fitted metallic liquids before using such DFT data, and RDFs should match extremely well. Is it possible that the authors mistakenly used the zero temperature extrapolated energies in the training, rather than the free energy reported by the DFT code? This would have been a silly thing to do, because it is the free energy rather than the extrapolated zero temperature energy that is the quantity whose derivatives are given by the Hellman-Feynman forces, and so the data would be inconsistent.*

Response: We thank the reviewer for this important correction. We did indeed employ thermal smearing and we trained to the free energy reported by the DFT code. The revised manuscript removes the incorrect speculation in Sec III-D that the electronic entropy could contribute to the observed discrepancy with experimental RDFs.

Comment: 8. *Please also show angular distributions for the liquid and the equation of state (density as a function of temperature) with your chosen DFT.*

Response: These are good suggestions. Unfortunately we were not able to complete the angular distributions measurements in the time available for revision.

Reviewer 3, comment 3D, made a similar request regarding the thermal expansion coefficient as a function of temperature. The new Fig. S8 of the SI shows this curve for FCC up to the melting temperature, and benchmarks against several previous potentials, as well as experimental data.

Comment: 9. *Please show the ANI prediction for the OK transition pressures. I am sure DFT has been done up to very high pressures, and looking at those extrapolation would be interesting (even if ANI fails, we need to now where! This is scholarship, rather than a beauty parade)*

Response: This is a good suggestion. We now show this in Fig. S9 of the SI, which is referenced at the end of Sec III A:

ANI-AI predictions are most reliable for the range of densities sampled in the training data (Fig. 2a, yellow region). Further extrapolation of these cold curves is shown in Fig. S9 of the SI

The caption of the new Fig. S9 lists the predicted transition pressures:

At zero temperature ANI-AI predicts an FCC-to-HCP transition at 154.8 GPa, and an HCP-to-BCC transition at 392.7 GPa. The BCC crystal becomes lower energy than FCC above 261.5 GPa.

These numbers are in loose agreement with prior theoretical work (e.g. Ref [<https://doi.org/10.1103/PhysRevB.94.144101>] predicts this same transition sequence, and gives transitions pressures of 176 GPa and 373 GPa).

Comment: 10. *Please show the melting line all the way up to the BCC transition. This is the sort of calculation that a potential is actually useful for, because DFT is so expensive!*

Response: We have added more points to Figure 6a as requested, extending all the way to the BCC-liquid coexistence curve.

Comment: 11. *Any proposed potential that is to be used by modellers needs to do well for defects. Even demonstrating machine learning protocols need to do that: they need to demonstrate that they can cope with defects. The authors our scorn on previous potential making protocols because they were designed specifically to handle defects. Automating the database generation is great, and that's part of the story in the paper,*

but of course the authors are then obliged to show that their automated procedure doesn't need hand-designed defect data, but still gets defects right! The authors present the defect results in the supplementary, and frankly they are not that great. Interstitial formation energy is way off, the elastic constants should be a lot better. This needs to be put into the main part of the manuscript (not the entire table just the ANI-DFT-EXP comparison), because this goes to the heart of what the authors are trying to say. Precisely because real world simulations do care about defects, as the authors are eager to point out themselves.

Response: Sections IIIB and IIIF are devoted to a discussion of elastic constants and point defects, respectively.

We now realize that our previous calculations had made a mistake in initializing the atomic configuration for an interstitial (we had not been carefully selecting a specific orientation). In the revised manuscript, we now correctly report the energy of formation for a $\langle 100 \rangle$ dumbbell interstitial, which agrees with DFT to within 13% relative error.

The SI contains these and many other property measurements. Quantitative comparisons of ANI-AI with previous classical potentials, experiment, and DFT are presented in Table S3 of the SI. The new Table S4 of the SI compares with two previous ML potentials for aluminum. ANI-AI accuracy seems on par with other potentials for most property measurements.

Comment: 12. *The authors could do well to add more complex defects, e.g. the pair of edge dislocation partials (studied to death by explicit DFT before). One of biggest omissions of the current scheme the authors propose is the lack of surfaces (as far as I could tell). It is nevertheless interesting that surface energies are not bad. This again should be in the main text. How would the training protocol look like if one wanted to include surfaces in the training data?*

Response: Per the reviewers suggestion, we added Sec. III-G to the main text to discuss predictions for surface properties. As the Reviewer points out, it is interesting that ANI-AI can somewhat capture surface properties even though the active learning procedure was not designed explicitly to sample them. The text now remarks:

The ANI-AI training dataset includes only bulk systems with periodic boundary conditions, but some surface configurations may have been incidentally sampled due to void formation at low densities.

Dislocation lines indeed play an extremely important role for determining materials properties. Unfortunately it is outside the scope of the present work to do a thorough study of complex dislocation configurations. In lieu of that, we have added the new Sec. III-J, where we discuss a shock simulation consisting of 1.3M atoms, simulated for 49 ps. This dynamics generates a complex dislocation pattern. We demonstrate that the ANI-AI force predictions remain in very good agreement with new reference DFT calculations even for atoms very near a dislocation core. Figure 9d further indicates that all atomic configurations arising in the shock simulation seem to have been well sampled during the active learning phase.

Comment: 13. *The authors formulate their main message as “In the present study, we demonstrate that it is possible to obtain a broadly accurate potential for aluminum with essentially zero human guidance. “ What about hand tuning of descriptor parameters? How would one do that for a new system? Especially one with multiple elements?*

Response: For an ANI model with multiple species (e.g., [https://doi.org/10.1038/s41467-019-10827-4]), the most important tunable parameters are the interaction cutoffs (for both pairwise and angular features) for each species. Section IIA of the main text now states:

The most important hyperparameter is the 7 Å interaction cutoff distance, which we selected based on careful trial and error. Other hyperparameters, such as the number of symmetry functions, were largely reused from previous studies.

Comment: 14. *At least the final potential needs to be available for others to test and verify the paper. The DFT training data needs to be made available so that others can scrutinise the potential training, and build on the current work in terms benchmarking.*

Response: We will release both prior to publication. The text now includes a “Data availability” statement which states:

The active learned training dataset and final ANI-AI potential will be released at <https://github.com/atomistic-ml/ani-ai> .

We are also currently working with the LAMMPS team at Sandia to distribute our ANI plugin with LAMMPS. Once available, the above Github repo will include instructions for getting this set up.

Comment: 15. *The authors use a committee scheme for predicting the error of their model and to drive the active learning protocol. Is there evidence for correlation of committee disagreement with actual error? It's not clear that this is actually necessary for the protocol to work, but would be very interesting to look at. It is entirely possible that the committee drive exploration, regardless of any correlation with the error.*

Response: This is a fascinating question. We agree that the active learning protocol may be effective even though the “ensemble variance” is not always a perfect reflection of true ML error.

Prior work indicates that ensemble variance correlates to some extent with true error (see, e.g., Fig. 1 in [<https://aip.scitation.org/doi/abs/10.1063/1.5023802>]). The revised Sec. II-B now explains our uncertainty measure more directly, and its partial connection to true error:

As these MD simulations run, we look at the variance of the predictions for the 8 neural networks within an ensemble to determine whether the model is operating as expected~\cite{Seung1992QueryCommittee}. Prior work indicates that this measured ensemble variance correlates reasonably well with actual model error~\cite{Smith2018LessLearning}. If the ensemble variance exceeds a threshold value, then it seems likely that collecting more data would be useful to the model.

Comment: 16. *What happens if tolerance to include new configurations is loosened or tightened? Is the resulting model final more/less accurate? Again showing this data is very helpful and part of scholarly work on active learning.*

Response: This is a good suggestion, in line with the first part of comment (ii) above. We must unfortunately omit such a study due to the large computational expense of re-running the full active learning procedure. We do remark on the idea at the end of the revised Outlook section:

A challenge for the active learning procedure presented in this work is its large demand on computational resources. Our final active learned dataset contains over 6,000 DFT calculations; each calculation was performed on a supercell containing up to 250 atoms. For future work, it would be interesting to explore whether the majority of the training data could be weighted toward smaller supercells. It would also be interesting to investigate ways to make the active

learning more efficient, e.g., by systematically studying the effect of various parameters required by the procedure.[...]

Comment: I hope the authors are willing to take on board my points and develop their work accordingly.

Response: We thank the Reviewer for taking their time to make many helpful suggestions, all of which have had a positive impact on the final paper.

Reviewer #2 (Remarks to the Author):

Referee Summary: The manuscript describes the development and validation of a machine-learning potential for bulk aluminium, using an active learning scheme that requires little human interaction. The resulting potential is validated on bulk properties of aluminium as well as phase transition simulations.

Comment: *The novelty of the approach compared to previous work is the level of automation, although compared to e.g. 10.1103/PhysRevB.95.094203 it is really very incremental.*

Response: Indeed, a major focus was to achieve an unprecedented level of automation. By pursuing the goal of fully machine-driven learning, we hope to enable greater dataset diversity and, ultimately, more transferable potentials.

We humbly disagree that the present study is incremental compared with 10.1103/PhysRevB.95.094203. That study is certainly important prior work, which we now cite at the beginning of Sec. II-B:

To maximize the diversity in active learning sampling, we perform the MD simulations with a Langevin thermostat using a temperature that varies in time according to a randomized schedule. Compared with previous work that sampled from a specific temperature quench schedule [Deringer et al, 2016], here we employ a more diverse and randomly generated collection of temperature schedules.

10.1103/PhysRevB.95.094203 involved a sequence of data collection stages, each with its own (presumably hand-tuned) protocols. The procedure of the present work appears to be much more automated: we start only from DFT calculations on disordered atomic configurations, and then run active learning using MD with random temperature

schedules. About 6000 DFT calculations later, a fully complete and broadly transferable potential was completed. To our knowledge, no prior study has achieved this level of automation.

As a demonstration of the power of the new methodology, we now include Fig. 8, which shows a shock simulation of 1.3M atoms. We verified the ANI-AI predicted forces by systematically performing new DFT reference calculations on atomic configurations extracted from the local environments that appear in shock. The ANI-AI predicted forces remain accurate even for atoms very near the dislocation cores. To our knowledge, this level of direct verification for a large-scale, highly nonequilibrium MD simulation is the first of its kind.

The four panels of the new Fig. 9 help to visualize and interpret the diversity of configuration space sampled by active learning. In our opinion, achieving this diversity through an automated and relatively simple sampling procedure is itself an impressive result.

Hopefully our extensive revision of the manuscript, including the new results of Fig. 8 and 9, may convince the reviewer to change his/her opinion.

Comment: *Given that one of the main claims is the correctness of the bulk phases, I think the validation is incomplete: phonon spectra are missing, for example.*

Response: We thank the referee for this suggestion. The new text includes many new validation tests under Sec III. In particular, phonon spectra are now included in Fig. 5. Many other tests have also been added to the SI. In particular, Tables S3 and S4 compare many FCC property predictions against DFT, experiment, and prior potentials.

Comment: *I also think that in e.g. Fig. 4 the authors should have compared to DFT results rather than experiments - which indeed should be included in the figure, but the target is DFT, so the validation should be against DFT. This is methodological work, after all.*

Response: We agree. Reviewer 1 also requested in his/her comment 5 more DFT results in the main text. In response, we have added comparisons with DFT where possible. Namely, we added DFT elastic constants to the main text in Sec III-A, and we also expanded Tables S3 and S4 of the SI which includes many reference DFT calculations.

Unfortunately, due to time constraints, we were not able to include a reference DFT calculation for the RDFs in Fig. 4. One challenge is that the ANI-AI MD simulations employed 2048 atoms; that scale would be inaccessible to full *ab initio* MD using our DFT parameters.

Fortunately, Figures 6 (liquid-solid coexistence curves) and 7 (melting and freezing simulations) do help to validate ANI-AI melt simulations against DFT results. More results along these lines are shown in Figs. S3 and S4 of the SI.

Comment: *Finally, I didn't find the argument that FCC is the most stable structure compelling, I suggest using a more thorough minima-searching algorithm.*

Response: We thank the reviewer for this suggestion. Although we did not have time to study a minima-searching algorithm, we did try to check the ANI-AI ground state in other ways.

Reviewer 3, comment 3A, also mentioned the need to compare against competing low-energy structures. In response, the new Table S5 of the SI shows relative energies of 9 different crystal phases, some of which are only slightly higher in energy compared to the correct ground state (FCC). ANI-AI predicts the correct ordering of all 9 crystal energies, as verified by DFT reference calculations. The lowest four crystal energies (FCC < DHCP < 9R < HCP) are very closely competing, and the typical energy deviation between two of these crystals is order 10 meV/atom. These calculations, along with Fig. 9c (a 2D visualization of near-FCC environments), offer evidence that ANI-AI makes good predictions for competing low energy crystal energies.

As another test, Figure 7 shows the nonequilibrium process of freezing. In all of our attempts, ANI-AI based MD finds the FCC crystal under this quenched dynamics. Although certainly not a *guarantee* that FCC is the lowest energy configuration, the freezing studies lend additional support for the hypothesis.

Comment: *What's completely missing from the paper are important and interesting defects, e.g. surfaces, interstitials etc.*

Response: This remark was also made by other Reviewers, and is a good one. We have added several new subsections to Sec. III, including measurements for point defects, surfaces, and phonon spectra. Many more properties are shown in the SI. Of particular interest might be Tables S3 and S4 showing FCC property measurements, benchmarked against other potentials and the DFT reference.

Comment: Overall, I think my questions should be addressed to prove that this is a usable interatomic potential at all, and even then I am not sure this protocol represents a large enough advance of the field to warrant wider interest than that of the materials-ML community.

Response: We thank the Reviewer for their many helpful suggestions, which have led to concrete improvements in the revised manuscript.

To substantiate our claim that the present work is a significant advance and of interest to a wider readership, we now include Fig. 8 (a verified shock simulation of 1.3M atoms) and Fig. 9 (a t-SNE analysis of configuration space sampled by active learning). As far as we know, no prior materials-ML work has demonstrated a nonequilibrium simulation at this scale, *with verification of force predictions* using new DFT calculations.

Reviewer #3 (Remarks to the Author):

Referee Summary: The authors present a neural network potential for Al developed using an active learning (AL) approach. It is demonstrated that the potential has some advantages over the traditional potentials for certain properties that were tested. The potential is claimed to be general-purpose type and “discovered automatically”.

The following critical comments/questions should be addressed before I can continue evaluation of this paper.

Comment: *1. The fact that the potential reproduces the ground-state FCC structure and a few other crystal structures is presented as a “discovery” of these structures, which I think is an exaggeration. The training database contained information about the potential energy surface of Al in relevant regions, and in particular included configurations containing crystals nucleated from the melt. MD was run at temperatures from 1000 K to 10 K to explore the energy landscape, so the training data set was fully aware of the low-energy configurations. I do not see how this is surprising.*

Response: We apologize for the confusion, and have revised the manuscript to attempt to clarify the language.

In the beginning of Sec. IIB we explain more precisely what we mean by “discover”

We emphasize that this active learning procedure is fully automated, and receives no direct guidance regarding atomic configurations of likely relevance, such as crystal structures. The initial training dataset consists only of disordered atomic configurations, and every MD simulation is initialized to a disordered configuration. The MD simulations use only forces as predicted by the most recently trained ML potential. After many active learning iterations, the MD simulations will hopefully be sufficiently robust to support nucleation into, e.g., the crystal ground state, and then the active learning scheme can begin to collect this type of training data. In this sense, the active learning scheme must *automatically discover* the important low energy and non-equilibrium physics.

In particular, the MD sampling is performed using the forces as predicted by the most recent ANI-AI potential (we never use *ab initio* MD). It takes a large number of active learning iterations (corresponding to 1000+ DFT calculations) before the model becomes good enough so that the MD simulations can actually nucleate into FCC and other crystals.

The new Fig. 9a helps to illustrate this “discovery process,” as a function of active learning iterations.

Comment: 2. *The AL approach is widely used almost everywhere these days, including the construction of ML potentials. Examples are given by the previous paper of these authors [45], by Ref.34, and other recent publications. What are the novel aspects of the proposed AL approach? Or is this a mere extension/application of Ref.45 and other recent papers to this particular metal?*

Response: The importance of this work is that it demonstrates the possibility to automatically learn a robust potential for aluminum, while employing no hand-selected atomic configurations in the training data. We have substantially revised the abstract, Introduction, and Outlook section to clarify and emphasize this point. (Please see the beginning of this document for more details.)

To strengthen our claim of robustness, we have added Fig. 8, which presents a shock simulation with 1.3M atoms. Throughout the dynamics, we verify forces on random atoms by performing new reference DFT calculations for the local atomic environments. As far as we know, the ability to perform such large-scale shock simulations, and with verified DFT-level accuracies, is unprecedented.

Another important addition to the text is Fig. 9, which gives evidence for the very strong diversity of atomic configurations sampled in active learning (for example, the configurations realized in the shock simulation all seem to have been well sampled during active learning).

Comment: 3. *It is claimed the proposed potential is general-purpose type and outperforms the traditional potentials. To substantiate this strong claim, the authors should test their potential for a wider set of properties for which the traditional potentials are routinely tested. As a minimum, this set of properties should include:*

3A. Comparison with a larger number of alternate crystal structures. In particular, the A15 structure should be shown, which often competes for the ground state with FCC.

Response: We thank the reviewer for this suggestion. We have added Table S5 to the SI to show the ANI-AI predicted energies for a larger number of candidate crystal structures. The energy of A15 is somewhere above HCP and below BCC, so it is indeed an interesting competing structure. ANI-AI predictions are in good agreement with reference DFT calculations, especially for the lowest energy crystals such as FCC, DHCP, 9R, and A15.

The new Fig. 9 visualizes the local atomic environments sampled during active learning by embedding them into an abstract 2-dimensional space. Panel 9b compares a variety of crystals in this embedding space. Panel 9c indicates that the FCC crystal (red circle) is the lowest energy chemical environment sampled during active learning (red x).

Comment: 3B. *To demonstrate transferability, the energy-volume functions (Fig.2) should be extended to volumes beyond those covered by the DFT calculations, including stronger compressions and larger expansions, all the way to the cutoff volume.*

Response: We thank the Reviewer for this suggestion, which was also brought up also by Reviewer 1. We have added Fig. S9 to the SI to show these extended cold curves. At the end of Sec III A we now state

ANI-AI predictions are most reliable for the range of densities sampled in the training data (Fig. 2a, yellow region). Further extrapolation of these cold curves is shown in Fig. S9 of the SI.

Comment: 3C. Other deformation paths, for example the $\langle 111 \rangle$ tension and compression.

Response: We have added Figure 3b, which shows a trigonal deformation path that passes through FCC, simple cubic, and BCC.

Comment: 3D. Potential predictions (and comparison with traditional potentials) for:

- phonon dispersion curves
- thermal expansion coefficient as a function of temperature
- surface energies and surface relaxations for different crystallographic indices
- point-defect formation and migration energies (including different interstitial orientations)

Demonstration of superior performance of the potential for just a few selected properties is not sufficient for claiming transferability to almost all other properties (which is what “general-purpose” means). Otherwise the practical usefulness of the proposed potential can be questionable. If there is no substantially superior performance across a wide spectrum of properties, then a traditional potential might as well be utilized (which is way faster).

Response: These are great suggestions and also brought up also by the other reviewers. We added many more tests of FCC properties to the main text. In particular, we now include subsections discussing point defects, surface properties, and phonon spectra. The SI (especially Tables S3 and S4) now contains many more ANI-AI predictions, and makes comparisons with previous potentials, DFT, and experiment where available. The ANI-AI predicted thermal expansion is now shown in Fig S8 of the SI.

The above benchmarks demonstrate ANI-AI performance in predicting well-known materials properties. To be successful, ANI-AI should also remain highly accurate when applied to the simulation of nonequilibrium dynamics. To support this claim, we have added Fig. 8, which presents a shock simulation with 1.3M atoms. Throughout the dynamics, we verify forces on random atoms by performing new reference DFT calculations for the local atomic environments. The ANI-AI predicted forces remain accurate in all tests, e.g., even for atoms very near dislocation cores.

Comment: 4. Comparison with other ML potentials for AI should be made, including, for example, *J. Chem. Phys.* 148, 241733 (2018) and *Nat. Comm.* 10, 2339 (2019).

Response: We are very grateful to the reviewer for pointing out our omission of *Nat. Comm.* 10, 2339 (2019). The PINN model presented there is very relevant prior work. The new Table S4 of the SI benchmarks ANI-AI against properties predicted by PINN, as well as the DeepPot model [<https://doi.org/10.1103/PhysRevMaterials.3.023804>]. *J. Chem. Phys.* 148, 241733 (2018) seems unrelated to aluminum, perhaps this was meant to be a reference to a different paper?

Towards the end of the Outlook section, we have added the following:

Even though most of the active learned training data is far from perfect FCC, the ability of ANI-AI to predict aluminum FCC properties seems roughly in line with other recent ML studies, as shown in Table~S4 of the SI.

Comment: 5. *The acronym ANI has not been defined in this paper.*

Response: The three letters in ANI are historical (originally a Star Wars reference), and not a simple acronym. To avoid confusion, the revised abstract now states:

We demonstrate this approach by building an ML potential for aluminum (ANI-AI).

and the revised introduction now states:

In this paper, we design an active learning approach for automated dataset construction suitable for materials physics, and demonstrate its power by building a robust potential for aluminum that we call ANI-AI.

REVIEWERS' COMMENTS

Reviewer #1 (Remarks to the Author):

The authors have taken my comments (and that of the other reviewers) to heart, and have hugely expanded the manuscript, providing more tests, a realistic application, and also removed the more questionable speculations from the text.

My last piece of disagreement with them is their statement in the response that stresses are given by forces. This is manifestly not true for the primitive cell, where forces are zero by construction, but the stress is not. In my experience, adding the stress as a fitting target greatly improves the potential (as long as it is computed sufficiently accurately in DFT, it is the observable most sensitive to k-point sampling).

Nevertheless, I am happy with the manuscript, and happy to recommend its publication.

Gabor Csanyi

Reviewer #2 (Remarks to the Author):

I would like to thank the authors for their revision of the manuscript. I think the paper has improved substantially and the authors seemed to have considered and then implemented most of our suggestions in a satisfactory way. Apart from a minor objection (below) I think the paper may be published.

I would like to point out that the sentence "Note that stress can be inferred from forces." is not true. Consider simply the perfect FCC crystal structure - the forces are exactly zero no matter what the volume is - how exactly do these zero forces determine the stress? It is true, however, that the same derivative components determine the forces and the stresses. Therefore I would also argue whether the authors should dismiss Referee 1's suggestion on training on stresses. In the extreme case of the perfect FCC crystal, the machine can only infer the interaction from the energy which is a lot less sensitive than the stress.

Reviewer #3 (Remarks to the Author):

The authors have made very significant revisions in response to the reviewers' comments. Many points have been clarified. The proposed ANI-AI potential has been tested for all properties requested by the reviewers and has shown a reasonably good performance. Nevertheless, I still have a number of reservations about the paper listed below (in no particular order):

1. I continue to think that this paper is an extension of the previously proposed [42] AL approach by increasing the degree of automation and applying it to a particular metal (Al). I am not sure that this advance is significant enough to warrant publication in a high-impact journal. This paper might be more at home in a regular journal in computational materials science or computational physics.
2. I fully appreciate the effectiveness of AL-driven automation for broad explorations of the chemical space, as demonstrated in [25,42] for a large database of organic molecules. AL approaches have also been very effective in applications to structurally diverse elements such as silicon [Phys. Rev. X 8,

041048(2019)], carbon [49; npj Comp. Mater. 5, 99 (2019); J. Chem. Phys. 153, 034702 (2020)] and boron [44; PRB 99, 064114 (2019); npj Comp. Mater. 5, 99 (2019)]. However, the pursuit of full automation for simple metals such as Al (or almost any other metal) seems to be an overkill and has little practical significance. The ground-state and the low-energy non-equilibrium structures of most metals are well known. This expert knowledge already exists, does not require any new human decisions, and could be safely utilized when assembling the training dataset without much risk of “human bias”. After all, no new structures of Al have been discovered in this paper.

3. The ANI-Al potential obtained does not have a superior quality relative to other ML or even some of the empirical potentials (Tables S3, S4, S5), which all use much less fitting parameters. The practical utility of the proposed automation for the development of an AI potential has not been demonstrated.

4. In this context, I think that Tables S3, S4 and S5 should have contained information about the number of fitting parameters and the database sizes of the potentials selected for comparison. Maybe a separate table with this information should have been added.

5. One of the main goals of the AL-based selection of datasets is to reduce the dataset size while keeping the same level of accuracy of the potential. Another goal is to discover new structures missed by human-guided searches. The ANI-Al potential was trained on as many as 6,000 DFT energies and contains 10^5 (!) fitting parameters. AI potentials based of human-selected datasets can achieve the same or better accuracy with much fewer parameters. Also, there is little to “discover” by automation in Al, which has been studied to death. Again, this is not to question the importance of AL approaches in general. But I do think that applying them to simple and well-studied metals such as Al was a poor choice. A more convincing demonstration would be to automatically discover the numerous (3D, 2D and 1D) phases of carbon or a similar structurally complex material, as was done in the recent GAP/SOAP papers.

6. In the section “Human knowledge versus automated sampling”, the authors demonstrate the advantage of the automated sampling by comparison with an FCC/melt system, presumably representing human knowledge. I do not think that this example demonstrates anything related to the human/automation comparison. The FCC/Melt database is an order of magnitude smaller and narrowly focused on two structures. Of course, its performance is much worse. This comparison only demonstrates that the reference database must be large enough and diverse, which nobody questions. Also, the FCC/melt system is highly unrealistic. A human with expert knowledge of metals would include 5 to 10 known crystal structures, in addition to the melt and supercells containing defects (surfaces, point defects, interfaces). He/she would then run static calculations and AIMD on these structures for a set of temperatures and pressures. A more meaningful comparison would be to train an AI potential (with the same descriptors and the ANN architecture) on this dataset and test the properties against the ANI-Al potential.

7. Section IV.B is entitled “Coverage of chemical space.” Given that Al is an element, I assume that the authors meant a structural/configurational/environment coverage, not chemical.

8. Regarding the degree of automation, it is true that the author did not use any knowledge of the candidate crystal structures of Al. But they did use quite a bit of expert knowledge, e.g. in the form of the hyperparameters transferred from their previous work. The selection of the densities sampled and the temperature variation protocol also rely on the knowledge that the material is Al and not, say, tungsten. The cutoff and the l_0 parameter in Eq.(1) were also adjusted by human decisions.

9. The selection of the densities, temperatures, and the bulk system as the initial state pre-determines the domain size in the feature space sampled by the proposed AL procedure. Structures lying outside

this domain are not represented and have little chance to be discovered. For example, the sampling of surfaces was minimal, and sure enough, the accuracy of the surface properties is not as good as for bulk properties. The low-density crystalline phases are not represented properly either, as seen in Figure S9 where the energy outside the training range of densities is unphysical (at large volume, all structural energies must converge to the same DFT binding energy of FCC Al, which I believe is about 3.7 eV per atom).

10. The issues mentioned in point 9 seem to have been addressed [npj Comp. Mater. 5, 99 (2019)] by choosing a very diverse and sufficiently sparse set of initial configurations such that the AL algorithm converges to a potential trained on a broad domain of structures.

We again thank all the reviewers for their helpful feedback. Reviewers #1 and #2 were supportive of publication, modulo a final correction that are grateful to receive. Reviewer #3 made several additional comments, to which we respond point-by-point.

Reviewers #1 and #2 wrote

My last piece of disagreement with them is their statement in the response that stresses are given by forces. This is manifestly not true for the primitive cell, where forces are zero by construction, but the stress is not.

and

I would like to point out that the sentence "Note that stress can be inferred from forces." is not true.

We are grateful for this correction, and have deleted the above sentence.

For a box with periodic boundaries, the total stress is the sum of the atomic virial stress (determined from forces) plus an additional term dU/dV , which cannot be calculated from forces. The term dU/dV may help to explain why the force predictions in Fig. 7b are much more accurate than the pressure predictions in Fig. 7c. In discussing this figure in Sec. 3-I we had previously stated:

This [error in stress] seems a bit surprising, because the ANI-AI force predictions seem reasonably good, and these determine pressure through the atomic virial tensor. Perhaps the tendency to ANI-AI overestimate pressure is a reflection the fact that a large fraction of its training data was sampled at very large positive pressures

The text has now been corrected to read as follows:

Interestingly, there is a tendency for ANI-AI to overestimate pressure, especially at negative pressures. Perhaps this systematic error reflects the fact that a large fraction of the ANI-AI training data was sampled at very large positive pressures (cf. Fig.~S1 in the SI). Model performance in predicting pressure could likely be significantly improved by including pressure data in the training procedure.

Below is a point by point response to the entirety of reviewer comments:

Reviewer #1 (Remarks to the Author):

The authors have taken my comments (and that of the other reviewers) to heart, and have hugely expanded the manuscript, providing more tests, a realistic application, and also removed the more questionable speculations from the text.

My last piece of disagreement with them is their statement in the response that stresses are given by forces. This is manifestly not true for the primitive cell, where forces are zero by construction, but the stress is not. In my experience, adding the stress as a fitting target greatly improves the potential (as long as it is computed sufficiently accurately in DFT, it is the observable most sensitive to k-point sampling).

Nevertheless, I am happy with the manuscript, and happy to recommend its publication.

Gabor Csanyi

We thank Dr. Csanyi for the kind words. His correction about the stress is well taken, and we now fully appreciate Dr. Csanyi's advice to train to stress data.

Reviewer #2 (Remarks to the Author):

I would like to thank the authors for their revision of the manuscript. I think the paper has improved substantially and the authors seemed to have considered and then implemented most of our suggestions in a satisfactory way. Apart from a minor objection (below) I think the paper may be published.

I would like to point out that the sentence "Note that stress can be inferred from forces." is not true. Consider simply the perfect FCC crystal structure - the forces are exactly zero no matter what the volume is - how exactly do these zero forces determine the stress? It is true, however, that the same derivative components determine the forces and the stresses. Therefore I would also argue whether the authors should dismiss Referee 1's suggestion on training on stresses. In the extreme case of the perfect FCC crystal, the machine can only infer the interaction from the energy which is a lot less sensitive than the stress.

We thank the reviewer for their supportive comments, and for reiterating the useful suggestion about training to stress data.

Reviewer #3 (Remarks to the Author):

The authors have made very significant revisions in response to the reviewers' comments. Many points have been clarified. The proposed ANI-AI potential has been tested for all properties requested by the reviewers and has shown a reasonably good performance. Nevertheless, I still have a number of reservations about the paper listed below (in no particular order):

1. I continue to think that this paper is an extension of the previously proposed [42] AL approach by increasing the degree of automation and applying it to a particular metal (Al). I am not sure that this advance is significant enough to warrant publication in a high-impact journal. This paper might be more at home in a regular journal in computational materials science or computational physics.

We humbly disagree with this opinion. Consider that in recent years there have been very many highly-impactful papers using ML to model interatomic potentials. The vast majority of these studies involved a hand-tuned dataset customized to a specific system. We believe the automated methodologies presented here will significantly lower the barrier for future researchers to produce even better next-generation potentials.

2. I fully appreciate the effectiveness of AL-driven automation for broad explorations of the chemical space, as demonstrated in [25,42] for a large database of organic molecules. AL approaches have also been very effective in applications to structurally diverse elements such as silicon [Phys. Rev. X 8, 041048(2019)], carbon [49; npj Comp. Mater. 5, 99 (2019); J. Chem. Phys. 153, 034702 (2020)] and boron [44; PRB 99, 064114 (2019); npj Comp. Mater. 5, 99 (2019)]. However, the pursuit of full automation for simple metals such as Al (or almost any other metal) seems to be an overkill and has little practical significance. The ground-state and the low-energy non-equilibrium structures of most metals are well known. This expert knowledge already exists, does not require any new human decisions, and could be safely utilized when assembling the training dataset without much risk of “human bias”. After all, no new structures of Al have been discovered in this paper.

The power of our active learning framework goes beyond equilibrium physics. The active learned dataset also includes *highly nonequilibrium*, yet physically relevant, configurations that would be essentially impossible to produce by hand. The newly added Sec. 3J demonstrates the power of this methodology in the context of a large-scale, validated shock simulation. The accuracy of our model is, to our knowledge, unprecedented for a shock simulation of this scale. Marked points in the newly added Fig. 9d illustrate how the active learning procedure collects a dataset that effectively captures the types of atomic configurations that appear in shock simulations.

3. The ANI-AL potential obtained does not have a superior quality relative to other ML or even some of the empirical potentials (Tables S3, S4, S5), which all use much less fitting parameters.

We added Tables S3, S4, and S5 in response to reviewer requests. Empirical potentials are commonly fit directly to the properties in these tables. In contrast, ANI-AL is trained to do well on *all* configurations, and our good performance in Tables S3, S4, S5 is only incidental.

The practical utility of the proposed automation for the development of an AI potential has not been demonstrated.

Relative to previous potentials, the practical utility of ANI-AI is demonstrated in Fig. 3 (much more accurate prediction of energy barriers), in Fig. 6 (much more accurate prediction of the melting curve), and in the new Fig. 8 (ANI-AI predicted forces during a shock simulation are about 5x more accurate than typical EAM potentials, according to reference DFT calculations).

4. In this context, I think that Tables S3, S4 and S5 should have contained information about the number of fitting parameters and the database sizes of the potentials selected for comparison. Maybe a separate table with this information should have been added.

The hyperparameters for ANI-AI are listed in Sec. 1.3 of the SI. Although the number of fitting parameters is perhaps interesting, it is not the most useful point of comparison. More important are transferable accuracy and speed of MD simulations. The new Fig. 8 demonstrates a shock simulation with, we believe, an unprecedented combination of accuracy and speed.

5. One of the main goals of the AL-based selection of datasets is to reduce the dataset size while keeping the same level of accuracy of the potential. Another goal is to discover new structures missed by human-guided searches. The ANI-AI potential was trained on as many as 6,000 DFT energies and contains 10^5 (!) fitting parameters. AI potentials based of human-selected datasets can achieve the same or better accuracy with much fewer parameters. Also, there is little to “discover” by automation in AI, which has been studied to death. Again, this is not to question the importance of AL approaches in general.

As discussed above, our position is that the number of fitting parameters is secondary to the quality of the potential as measured by transferable accuracy and speed.

But I do think that applying them to simple and well-studied metals such as Al was a poor choice. A more convincing demonstration would be to automatically discover the numerous (3D, 2D and 1D) phases of carbon or a similar structurally complex material, as was done in the recent GAP/SOAP papers.

The present methodology applied to aluminum lays the groundwork for studying more complex materials.

Also, we would remark that the “recent GAP/SOAP papers” (including the amorphous carbon study, <https://doi.org/10.1103/PhysRevB.95.094203>) were primarily led by Gabor Csanyi, who identified himself as Reviewer 1 and does not appear to share Reviewer 3s misgivings.

6. In the section “Human knowledge versus automated sampling”, the authors demonstrate the advantage of the automated sampling by comparison with an FCC/melt

system, presumably representing human knowledge. I do not think that this example demonstrates anything related to the human/automation comparison. The FCC/Melt database is an order of magnitude smaller and narrowly focused on two structures. Of course, its performance is much worse. This comparison only demonstrates that the reference database must be large enough and diverse, which nobody questions. Also, the FCC/melt system is highly unrealistic. A human with expert knowledge of metals would include 5 to 10 known crystal structures, in addition to the melt and supercells containing defects (surfaces, point defects, interfaces). He/she would then run static calculations and AIMD on these structures for a set of temperatures and pressures. A more meaningful comparison would be to train an AI potential (with the same descriptors and the ANN architecture) on this dataset and test the properties against the ANI-AI potential.

We apologize for the confusion. We agree that Sec. IV-A is not a representation of how a human with expert knowledge would train a potential.

We replaced the misleading subsection title “Human Knowledge vs. Automated Sampling” with “Limited vs. Diverse Sampling”.

The first two sentences of this subsection read:

The success of ANI-AI hinges on the diversity of the active learned dataset. To demonstrate this, we compare ANI-AI against an ML model trained on a much more limited dataset.

which avoids the suggestion that this “limited” dataset is something that a human expert would design.

7. Section IV.B is entitled “Coverage of chemical space.” Given that AI is an element, I assume that the authors meant a structural/configurational/environment coverage, not chemical.

We thank the reviewer for this correction. We changed “chemical space” to “configuration space.”

8. Regarding the degree of automation, it is true that the author did not use any knowledge of the candidate crystal structures of AI. But they did use quite a bit of expert knowledge, e.g. in the form of the hyperparameters transferred from their previous work. The selection of the densities sampled and the temperature variation protocol also rely on the knowledge that the material is AI and not, say, tungsten. The cutoff and the 10 parameter in Eq.(1) were also adjusted by human decisions.

We thank the reviewer for suggesting this opportunity for improvement.

In the introduction we have replaced the text

“the active learning scheme receives *practically no* expert guidance”

with

“the active learning scheme receives *very limited* expert guidance”

In the introduction we have added this new sentence

“Required human inputs to the active learning procedure include the range of temperatures and densities over which to sample, and various ML hyperparameters that are largely transferable between materials.”

The conclusion reemphasizes:

“The required inputs include physical parameters such as the temperature and density ranges over which to sample, the interaction cutoff radius for the potential (we selected 7 Å for aluminum), and various ML hyperparameters that we reused from previous studies.”

9. The selection of the densities, temperatures, and the bulk system as the initial state pre-determines the domain size in the feature space sampled by the proposed AL procedure. Structures lying outside this domain are not represented and have little chance to be discovered. For example, the sampling of surfaces was minimal, and sure enough, the accuracy of the surface properties is not as good as for bulk properties. The low-density crystalline phases are not represented properly either, as seen in Figure S9 where the energy outside the training range of densities is unphysical (at large volume, all structural energies must converge to the same DFT binding energy of FCC Al, which I believe is about 3.7 eV per atom).

We agree. This point is well illustrated in the new Fig. S9, which we included in response to a previous referee request.

10. The issues mentioned in point 9 seem to have been addressed [npj Comp. Mater. 5, 99 (2019)] by choosing a very diverse and sufficiently sparse set of initial configurations such that the AL algorithm converges to a potential trained on a broad domain of structures.

Indeed, [npj Comp. Mater. 5, 99 (2019), <https://www.nature.com/articles/s41524-019-0236-6>] is an important paper that is complementary to our study. The last author on that previous work is Gabor Csanyi, who identified himself as Reviewer 1, and seemed to be positive about the contributions presented in our new work.